# A strength inversion origin for non-volcanic tremor

Paola Vannucchi [1], Alexander Clarke[2], Albert de Montserrat [3], Audrey Ougier-Simonin[4], Luca Aldega [5] &
Jason P. Morgan [6✉]

Non-volcanic tremor is a particularly enigmatic form of seismic activity. In its most studied subduction zone setting, tremor typically occurs within the plate interface at or near the shallow and deep edges of the interseismically locked zone. Detailed seismic observations have shown that tremor is composed of repeating small low-frequency earthquakes, often accompanied by very-low-frequency earthquakes, all involving shear failure and slip. However, low-frequency earthquakes and very-low-frequency earthquakes within each cluster show nearly constant source durations for all observed magnitudes, which implies characteristic tremor sub-event sources of near-constant size. Here we integrate geological observations and geomechanical lab measurements on heterogeneous rock assemblages representative of the shallow tremor region offshore the Middle America Trench with numerical simulations to demonstrate that these tremor events are consistent with the seismic failure of relatively weaker blocks within a stronger matrix. In these subducting rocks, hydrothermalism has led to a strength-inversion from a weak matrix with relatively stronger blocks to a stronger matrix with embedded relatively weaker blocks. Tremor naturally occurs as the now-weaker blocks fail seismically while their surrounding matrix has not yet reached a state of general seismic failure.

[1] Università degli Studi di Firenze, Firenze 50121, Italy. [2] University College London, Gower Street, London WC1E 6BT, UK. [3] Università degli Studi di Padova, Padova 35131, Italy. [4] British Geological Survey, Keyworth NG12 5GG, UK. [5] Università di Roma, La Sapienza, Roma 00185, Italy. [6] SUSTech, Shenzhen 518000, China. ✉email: jason@sustech.edu.cn

Plate boundary faults can accommodate displacement in a spectrum of slip rates ranging from aseismic, continuous creep at plate motion speeds (mm/yr) to earthquakes slip at rates of ~1 m/s[1–3]. Intermediate between these end-members, non-volcanic tremor and slow slip events (SSE) occur episodically along most subduction plate interfaces as part of a slow earthquake process[1,4].

Subduction zone tremor is a long-duration (minutes to days) and low-amplitude seismic signal formed of repetitive subevents of low-frequency earthquakes (LFEs) of $M_w \leq 2$[5–7] and very-low-frequency earthquakes (VLFs) of $2 \leq M_w \leq 5$[6,8]. Tremor is associated with seismic shear failure like regular earthquakes[5,9]. Unlike earthquakes, LFEs and VLFs often do not fit the characteristic self-similar seismic moment-rate function of other types of seismicity; instead they have nearly constant source durations (Methods) over an event size range of two orders of magnitude[6–8]. The near-constant <1 s duration of LFEs ("Methods") implies seismic sources on the order of ~20–100 s metres in size[8–10], that fail with small but variable stress-drops ("Methods"). Supino et al.[9] suggest that the characteristic source size may be further constrained to ~80–350 m based on LFE moment duration scaling.

In general, deep tremor is nearly continuously distributed within a narrow, belt-like zone downdip of the interseismically locked zone[11] (Fig. 1). In the Nankai Trough, SW Japan, shallow tremor, instead, has a measured spatial pattern of discontinuous clusters located updip of the interseismically locked zone[11] (Fig. 1), possibly also involving upper plate faults[12]. In Costa Rica the tremor distribution[13,14] is apparently continuous from shallow to deep, and both tremor and SSEs[14,15] overlap with the region of megathrust earthquake rupture[15]. Tremor usually occurs more frequently during SSEs, but remains active at a lower rate during the time intervals between SSEs (inset in Fig. 1a).

Seismological observations depict a fault surface where tremor is produced by repetitive failure of asperities that are closely spaced, similar in size, yet not necessarily uniformly distributed[10]. The existence of mechanical heterogeneity along a fault surface has been usually invoked to explain tremor, with geologically observed fault-zone heterogeneity used to provide the conceptual structural framework. It remains unclear whether heterogeneity controls spatial variations in rheology/asperities[16], strain rate[17], along-fault fluid pressure[18], or combinations of these.

Here we show that a specific rheological relationship among the rock components within a fault zone—i.e., weaker blocks dispersed in a stronger matrix—can lead to shear failure that generates tectonic tremor. To do this, we combine field and microstructural observations of exhumed subduction plate boundaries representative of the shallow conditions of tremor occurrence, laboratory experiments to constrain rock strengths, and numerical models built to reproduce the heterogeneity and the deformation conditions along the plate boundary. This mechanism for tremor can be extrapolated to deeper megathrust environment where tremor also occurs in the region where a fluid-assisted basalt-to-eclogite phase transition seems likely to happen.

## Results

### The conventional model of subduction channel shear zones is stronger blocks in weaker matrix.
At exhumed subduction plate boundaries the plate interface is not a single planar fault. Instead it is a ~100–1000 m thick shear zone[19]. Paleo-plate boundary shear zones are commonly characterised by the occurrence of subduction channel mélanges, heterogeneous mixtures of rocks with a variety of compositions, and diagenetic/metamorphic grades[20]. Tremor within this plate interface would occur not in

the mode of failing strong spots along a discrete fault surface, but in tremorgenic sub-volumes within a finite thickness shear zone. At the San Andreas fault, LFEs and VLFs do not appear to lie along a single fault, but rather within a seismically well resolved 2 km-thick tremorgenic zone[21].

In the conventional block-in-matrix model strong and competent blocks are embedded in a weak and less competent matrix[16,18,22,23]. Note that in geology, *strength* is usually defined as the resistance to permanent deformation by either flow or fracture[24] (see Table S1 for common rheological terms in geology, and their corresponding terminology in continuum and fracture mechanics). The conventional block-in-matrix model predicts that brittle failure would produce a mesh structure[18] with blocks prone to tensile and hybrid failure, while the matrix—usually clay-rich—experiences distributed deformation or shear failure depending on strain rate[12]. Block failure was proposed to happen when blocks, exceeding ~50% of a mélange volume, clog the shear zone (clog meaning that strong blocks locally accumulate to form a strong mechanical structure that completely spans the shear zone). Clogging could amplify stresses within these blocks by up to a factor of fourteen[25], while reducing stresses in the matrix, and increasing the overall resistance to channel shear[25]. Block failure would be associated with tremor events and periods without block clogging and decreased shear resistance associated with SSEs[25]. This hypothesis based on prior numerical modelling[25] appears incompatible with the first-order observation that periods of enhanced tremor are associated with transient ~fortnight-long SSEs which involve strain-rates faster than the background rate of tectonic strain accumulation [4], with periods of enhanced tremor not being associated with the non SSE clogging periods as predicted by the block clogging hypothesis.

When strong blocks form less than 25% of a mélange volume, heterogeneous mixture theory [cf. [26]] shows that the stress they experience will be determined by the lower competence or viscosity of their surrounding matrix. In this case, the shear stress magnitude $\tau$ would be governed by the viscosity $\mu$ of the matrix and the velocity gradient within the channel $\delta V/\delta h$, with $V$ being the shear velocity and $h$ the thickness of the subduction channel. The magnitude estimated for the characteristic viscosity of a subduction channel is $\mu = 10^{18}$ Pa s[20], which also corresponds to typical values estimated for the lower crust in regions of lower crustal flow[27]. With this viscosity, a 100 m-thick subduction shear zone sheared at 10 mm/yr would have a shear stress $\tau = \mu(\delta V/\delta h)$ equals ≈3 MPa, too low to yield crustal material which usually possesses a yield strength >3 MPa[28], while a channel viscosity $\mu = 10^{17}$ Pa s or channel thickness of ~1 km would be associated with 10-fold lower shear stresses of order 0.3 MPa.

Figure 2a shows an example of this behaviour in a numerical model that simulates yielding in a shear zone with a more realistic viscoelastoplastic rock rheology ("Methods"). The shear zone is still 100 m-thick and sheared at 10 mm/yr. The matrix has a viscosity of $10^{18}$ Pa s and a cohesion $c = 20$ MPa ("Methods"). The embedded blocks have the same cohesion, but a higher viscosity of $10^{20}$ Pa s that suppresses their viscous deformation, so they deform as elastoplastic materials, where this type of plastic deformation is a proxy for brittle slip on fault surfaces (Methods). The numerical model shows no failure in either the strong blocks or their surrounding lower-viscosity matrix, which instead deforms by ductile creep (Model 1a - Fig. 2a). More examples are shown in the Supplementary Information. If the blocks are more cohesive than the matrix, and the matrix viscosity is high enough so that the resulting higher ambient stresses cause it to fail brittly (Model 1b - Fig. 2b), then the matrix develops concentrated shear bands that terminate at contacts with more cohesive blocks. This mode of deformation leads to shear patterns in the matrix consistent

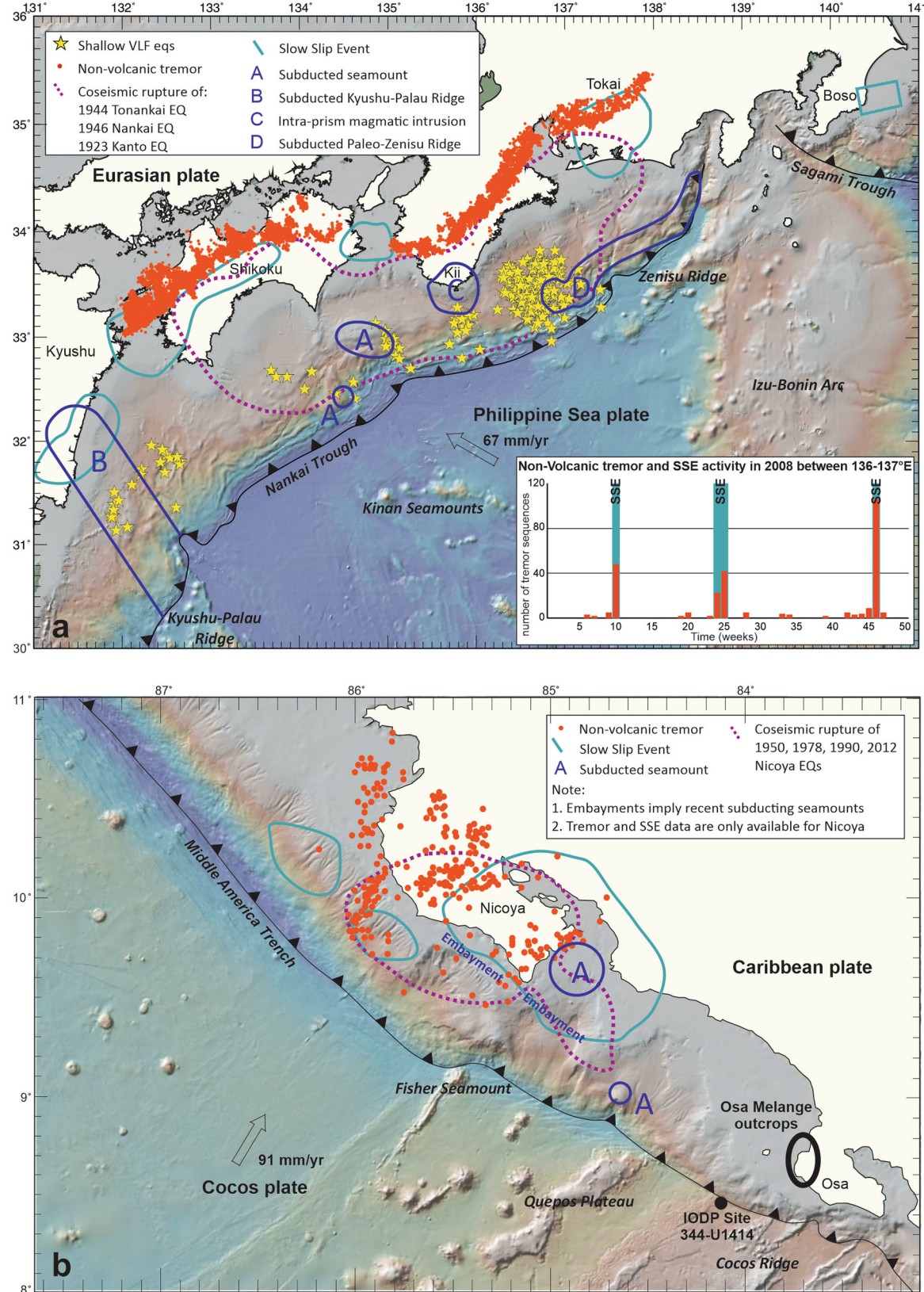

with a scaly-fabric[29] (Fig. 2b) or focused shear that ends at blocks (Fig. 2c) as predicted in the conventional block-in-matrix model[18], but here the blocks do not fail. These experiments generate deformation patterns that resemble those described in stronger block in weaker matrix mélanges with a low cohesion pelitic or metapelitic matrix[30–32], but with the implication that block failure in these types of mélanges is not tremorgenic. Indeed, models 1A–C (Fig. 2a–c) indicate that the deformation of strong tough blocks in a weaker and less tough matrix will not develop failure modes that have the

**Fig. 1 Distribution of slip modes and related tectonic structures in Nankai Trough and Costa Rica. a** Distribution of slow earthquakes along the Nankai Trough in SW Japan showing tectonic tremor distinguished as shallow very low frequency earthquakes (VLFs) and deep tremor[11], co-seismic megathrust rupture[2] and slow-slip events (SSEs)[2] compared to the location of subducted seamounts and magmatic bodies intruded within the accretionary prism[56,66], the subducted Kyushu Ridge[67], and the subducted paleo-Zenisu Ridge[66]. Inset: Number of non-volcanic tremors and occurrence of SSEs in 2008 between 136°E and 137°E longitude (from: "Slow Earthquake Database" http://www-solid.eps.s.u-tokyo.ac.jp/~sloweq/[68]). One degree of latitude is 111.11 km. **b** Distribution of slow earthquakes offshore Costa Rica compared to subducted seamounts[69,70]. Embayment in the forearc is assumed to be left by subducted seamounts[69]. Seismic and GPS data are here limited to the Nicoya Peninsula. The location of the outcrops of Osa Mélange referred in the text as well as IODP Site 344-U1414 are also shown in the map. One degree of latitude is 111.11 km.

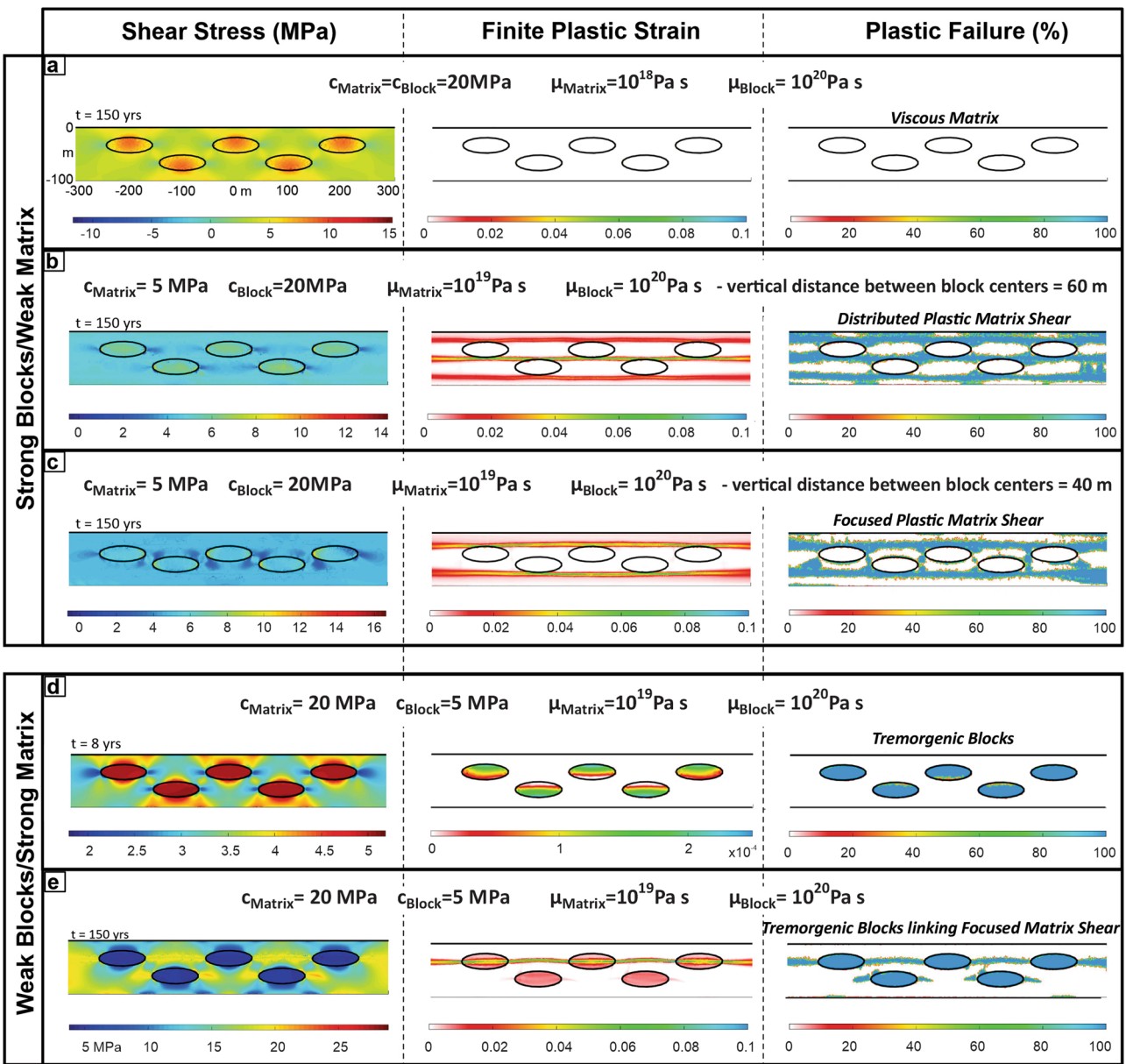

**Fig. 2 Numerical experiments on matrix and block deformation.** Shear channel 100 m wide and 1 km long; each panel shows the central 600 m. The overall shear rate across the channel is assumed to be 10 mm/yr with the top moving to the right. Both matrix and blocks are treated as variable visco-elasto-plastic materials. Initial model configuration and boundary conditions are given in the Methods section. **a** When the matrix is tough (cohesion $c = 20$ MPa) and lower-viscosity (viscosity $\mu = 10^{18}$ Pa s) while the blocks are tough ($c = 20$ MPa) and strong ($\mu = 10^{20}$ Pa s), the matrix will creep and neither blocks nor the matrix fail in a brittle/seismic mode. **b** When the matrix is less cohesive ($c = 5$ MPa) yet viscous enough ($\mu = 10^{19}$ Pa s) for ambient channel stresses to exceed its brittle yield strength while block properties are as in panel (**a**), the matrix can fail in a distributed mode. **c** When block and matrix properties are as in panel b, but the distance between blocks decreases, focused shear bands can form. d. When the block cohesion ($c = 5$ MPa) is 4-times lower than the matrix cohesion ($c = 20$ MPa), with block and matrix viscosities identical to the scenario of panel (**a**), the blocks start early ($t = 8$ yrs) to repeatedly fail in a tremorgenic mode, while the matrix creeps aseismically. **e** For the situation in d, but when the matrix viscosity is $10^{19}$ Pa s, channel stresses are higher. In this case, repeated block failure causes stresses in the matrix to build around the failing block's tips and induce high-stress-drop/seismic matrix failure in addition to continual low-stress-drop/tremorgenic block failure.

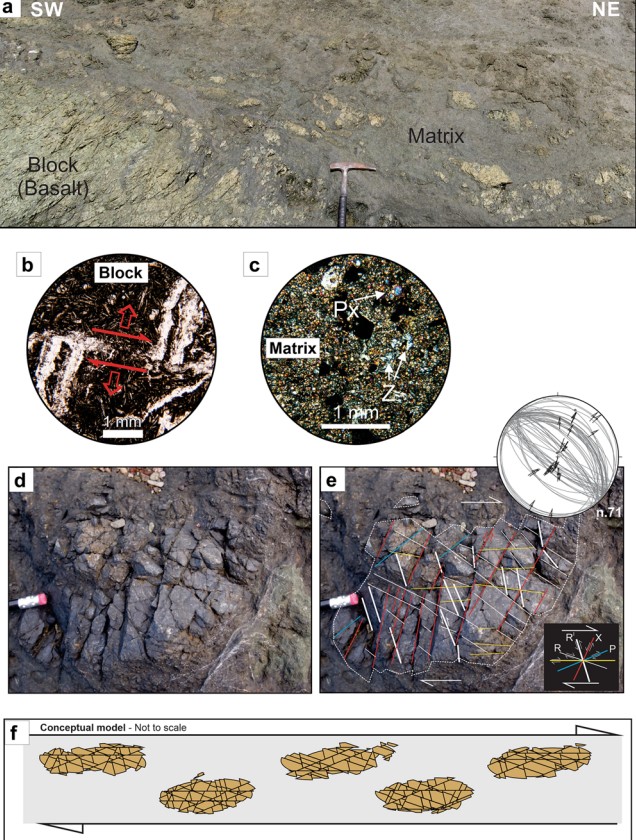

**Fig. 3 Osa Melange characteristic deformation. a** Typical block-in-matrix texture of the Osa Melange near Drake's Bay (Fig. 1b). **b** Petrographic photomicrograph in plane-polarised light of a basalt block showing a displaced calcite vein. **c** Petrographic photomicrograph of the matrix in cross-polarised light showing pyroxenes (Px) and zeolites (Z). **d** A block cut by a microfault network. **e** Trace and stereoplot (equal area lower hemisphere) of the microfaults showing Riedel-type geometry within a dextral shear zone. The jagged edges of the blocks are continuously sheared contributing to the progressive decrease of the blocks size. **f** Conceptual model showing how multiple generation of microfaults is ultimately creating a mosaic texture of the brecciated blocks. Here the blocks are cut by overprinted generations of Riedel systems with the geometry shown in the inset of Fig. 3e.

characteristic tens of metres failure size and variable yield-stress characteristics of observed seismic tremor.

**Weaker blocks in stronger matrix**. Field observations suggest that the assumption that strong/tough blocks are embedded in a weaker matrix is not always true. For example, in Costa Rica, where the plate boundary exhibits both shallow SSEs and tremor[15] (Fig. 1b), a mélange in the exhumed forearc in the NW of the Osa Peninsula shows clear field evidence that its blocks, initially stronger than their surrounding matrix, became weaker than the evolving matrix. Rock mechanic measurements of the frictional strength of block and matrix material (Fig. 3) further verify this strength contrast.

The Osa Mélange formed as an oceanic mass-wasting deposit related to seamount slope instabilities[33]. It contains igneous and sedimentary (mainly sand/siltstone, carbonate, chert) blocks, with sizes ranging from $10^{-2}$ to $10^2$ m and aspect ratios from 1:1 to 7:1, embedded in a volcanoclastic matrix (Fig. 3a). (See Supplementary Information for a more detailed description of the geological setting and composition of this mélange).

The mélange has been variably affected by hydrothermal alteration not exceeding the prehnite-pumpellyite facies (i.e., max $T \sim 250\,°C)^{34}$ with common smectite, zeolite, and calcite precipitation (Fig. 3b, c; Supplementary Information). The presence of smectite implies that the mélange did not experience temperatures associated with a depth of burial that activated the diagenetic transformation of smectite to mixed layer illite-smectite or chlorite-smectite (i.e., ~70 °C)[35].

The blocks in this mélange are pervasively brecciated by multiple sets of microfault systems resembling Riedel-type systems (Fig. 3d–f), often with several overprinted generations. In Fig. 3d, f widely spaced, evenly distributed microfaults forming primary Riedel shears (R) are displaced by succeeding secondary shorter R, R', and P shear surfaces while also reactivating some of the older R surfaces. These sets cross-cut each other nearly orthogonally. Shear fractures often have an extensional component (Fig. 3b) which suggests failure occurred under low effective normal stress[36] and low differential stress[37]. Breccia clasts have low internal deformation and often form a mosaic texture where shear is only detectable at the microscopic scale (Fig. 3b); in places comminution can develop depending on the amount of shear accommodated between clasts. This breccia is cohesive and variably sealed by mineral precipitation. Moreover, brecciated blocks often have diffuse boundaries that transition into the mélange matrix (Fig. 3f). The matrix is not foliated and only locally are shear structures coeval to block failure (Fig. 3a). Instead, the matrix is often pervasively cross-cut by arrays of planar to randomly oriented extension fractures and veins which indicate that fracturing occurred under varying differential stress conditions[37]. This fault/fracture mesh pattern—predominant shear failure in the blocks, smaller amounts of predominantly extensional failure in the matrix—implies that the blocks and the matrix did not have the same material properties and that the blocks were weaker than the matrix when they failed.

To test this interpretation, we performed triaxial frictional strength experiments on samples of basaltic blocks and volcanoclastic matrix. Experiments were conducted at room humidity conditions, confining pressure ($P_c$) varying from 60 to 120 MPa, and temperature ($T$) varying from 60 to 120 °C ("Methods"). The experiments show that the volcanoclastic matrix is between 3 and 10 times stronger in shear failure than its embedded basaltic blocks (Fig. 4 and "Methods").

To further test whether the strength inversion arises because of syn/post-subduction weakening of the blocks, we also conducted triaxial experiments on basalts recovered from the Cocos Ridge at IODP (International Ocean Discovery Programme) Site U1414 during Expedition 344[38]. This test assumes that these are the modern analogue to Osa Mélange basalts[33]. They indeed have similar phenocrysts and groundmass composition as well as having also experienced hydrothermal alteration temperatures from 170 to 220 °C[39]. Figure 4 shows that although the Cocos Ridge basalt samples can be up to twice as strong in shear failure than the mélange basaltic blocks, they are still ~3–4 times weaker in shear failure than the volcanoclastic matrix of the mélange. The strength inversion between blocks and matrix in the Osa Mélange is, therefore, primarily the product of the relative strengthening of the matrix. The Osa Mélange also contains field evidence that the volcanoclastic matrix was initially weaker than the basaltic blocks. Block boundaries sometimes preserve matrix injection-related structures such as sediment-filled fractures[33], implying that there was an early deformation stage when the matrix was able to preferentially flow during bulk deformation. Interestingly the Cocos Ridge basalts were covered by 70 m of strongly lithified volcanoclastic sandstones[38]. Lithification of these sediments has been related to hot advective fluids linked to Galapagos hotspot activity[39]. The mélange's matrix strengthening could be due to

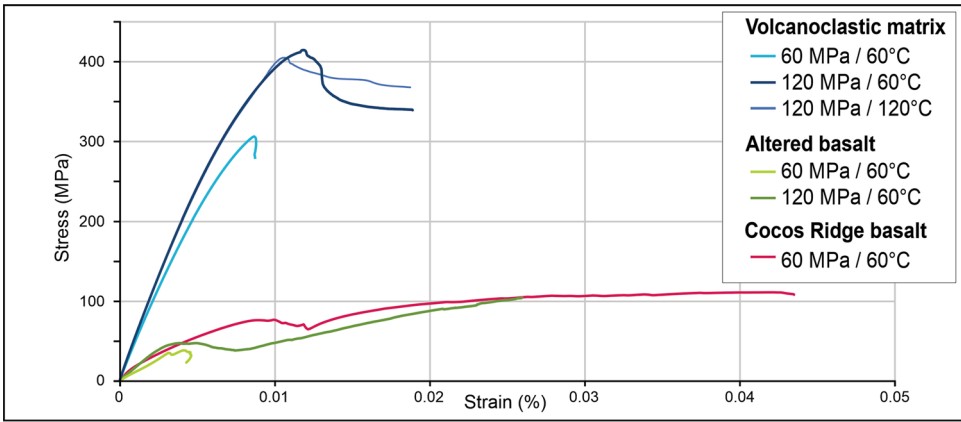

**Fig. 4 Triaxial strength experiments.** Triaxial experiment results for the basalt blocks and volcanoclastic matrix of the Osa Melange, and the Cocos Ridge basalt sampled during IODP Expedition 344 at Site U1414.

processes such as cementation by zeolite-producing reactions[40] and/or sediment load[41]. In an analogous setting offshore Hawaii, heat-flow measurements imply a phase of long-lived hydro-thermal activity within sediments created by mass wasting from the volcanic island chain[42]. Based on this information, we explored another set of numerical models in which the blocks are still more viscous, but now less tough (less cohesive) than the matrix, for example $\mu_{\mathrm{BLOCKS}} = 10^{20}$ Pa s, $c_{\mathrm{BLOCKS}} = 5$ MPa, and $\mu_{\mathrm{MATRIX}} = 10^{18}$ Pa s, $c_{\mathrm{MATRIX}} = 20$ MPa. Even though the blocks are still higher viscosity than the matrix, in this case, shear leads to repeated brittle failure of the blocks (Fig. 2d). More examples are shown in the Supplementary Information. In these models, the effect of the reduced cohesion of the blocks is analogous to what would happen if the blocks alone had an increase in their local fluid pressure. Failure initiates via yielding in the blocks located in the upper part of the shear zone—i.e., closest to the sliding boundary—and evolves so that different parts of the blocks progressively break and slip. Eventually, the blocks in the lower part of the shear zone also fail. In a similar experiment with a higher matrix viscosity, after blocks fail, the matrix becomes stressed enough so that further bulk shearing induces matrix failure around the tips of the deforming blocks where stresses concentrate (Fig. 2e). Matrix stresses continue to concentrate in this region until matrix failure bridges the gap between adjacent blocks. At this point, small slip events in the blocks and in the adjacent matrix are not able to sufficiently relieve stresses in the bulk of the matrix, so that continued shear activates failure in a wider, but still focused zone linking blocks and matrix (Figs. 2e and 5). These models are representative of the general situation of shearing of a block-in-matrix fabric irrespective of whether the fabric originated from mass wasting, as in the Osa Mélange, or from boudinage as in other settings[32,43–45].

## Discussion

We suggest that tectonic tremor is due to frequent small failure events that occur in relatively weak blocks. The resulting stress build-up between two failing weaker blocks (Fig. 5) could trigger local rupture of the matrix associated with the higher yield stresses and stress drops typical of micro-earthquakes and earthquakes. These higher stresses would also tend to reduce the viscosity of the matrix, if it follows a power-law creep rheology, which could lead to higher matrix creep-rates/SSEs. In this case, periods of slow slip, enhanced channel shear-rates, and enhanced tremor would be predicted to coexist, instead of occurring out of phase as implied by the conventional strong block failure scenario for tremor[25]. Furthermore, if stronger blocks fail when a clump of blocks clogs the shear channel, it is difficult to see how tremor

(e.g., individual block failure events) and larger seismic failure events could coexist as observed[12]. For example, if a seismic event is due to a cascade of block-failure events, it should suppress tremor events along its portion of the shear channel. In contrast, failure of weaker blocks would result in swarms of low-magnitude LFEs and VLFs, while the resulting stress build-up in the matrix between two failing weaker blocks could trigger local rupture of the matrix associated with micro-earthquakes as observed[12].

In subduction systems, tremor, and micro-earthquakes differ in their stress-drops: higher for micro-earthquakes than for tremor[46] ("Methods"). This observed behaviour is consistent with a weaker blocks in stronger matrix scenario as seen in the numerical experiment shown in Fig. 2e. Specifically, the lower bound of ~0.03 MPa for the stress-drop of an LFE placed by the stresses associated with both passing teleseismic waves[47] and tides[48] constrains the diameter of the LFE slip patch to be ~20–400 m (Methods). It further implies that the rupture speed of these events is ~100–400 m/s, considerably slower than the typical rupture speeds (1200–4000 m/s) of normal seismic events[49] and also indicates that the stress-drop associated with an individual LFE event is ~0.03–0.4 MPa, much lower than the ~4 MPa median stress-drop of a typical earthquake[50]. In a subduction channel characterised by a weak block-in-strong matrix fabric, the stress drop associated with failure of blocks is approximately given by $\Delta\tau \approx Gd/W$ where $G$ is the elastic shear modulus of the block, $d$ is the amount of slip across the block, and $W$ is the down-dip block width/thickness. In a shear zone with a down dip uniform velocity gradient, the amount of slip across the block will be a function of the block width, therefore $d$ and $W$ are pro-portional, and the ratio $d/W$ is approximately independent of block size. The effective shear modulus decreases either by frac-turing or when fluid pressure increases. High fluid pressure also reduces differential stress[51] and is thought to reduce stress drop during seismic failure[52]. The characteristics of weak block brec-ciation in Osa described above indicate high to lithostatic fluid pressure, more consistent with tremor than micro-earthquake events.

The proposed rheological strength inversion could be created along a subduction zone in different ways (Fig. 6), for a broad spectrum of matrix and block compositions. In this paper we focused on a mélange with altered basalt blocks and a matrix formed by volcanoclastic sediments likely related to a seamount apron[33], but which also constitute a common arc-sourced com-ponent of the subducting plate's sediments[53]. Prior to subduction, hydrothermal alteration could lead to patches of local matrix strengthening where tremor could later occur. In the case of subducted deposits related to seamount aprons, tremor should

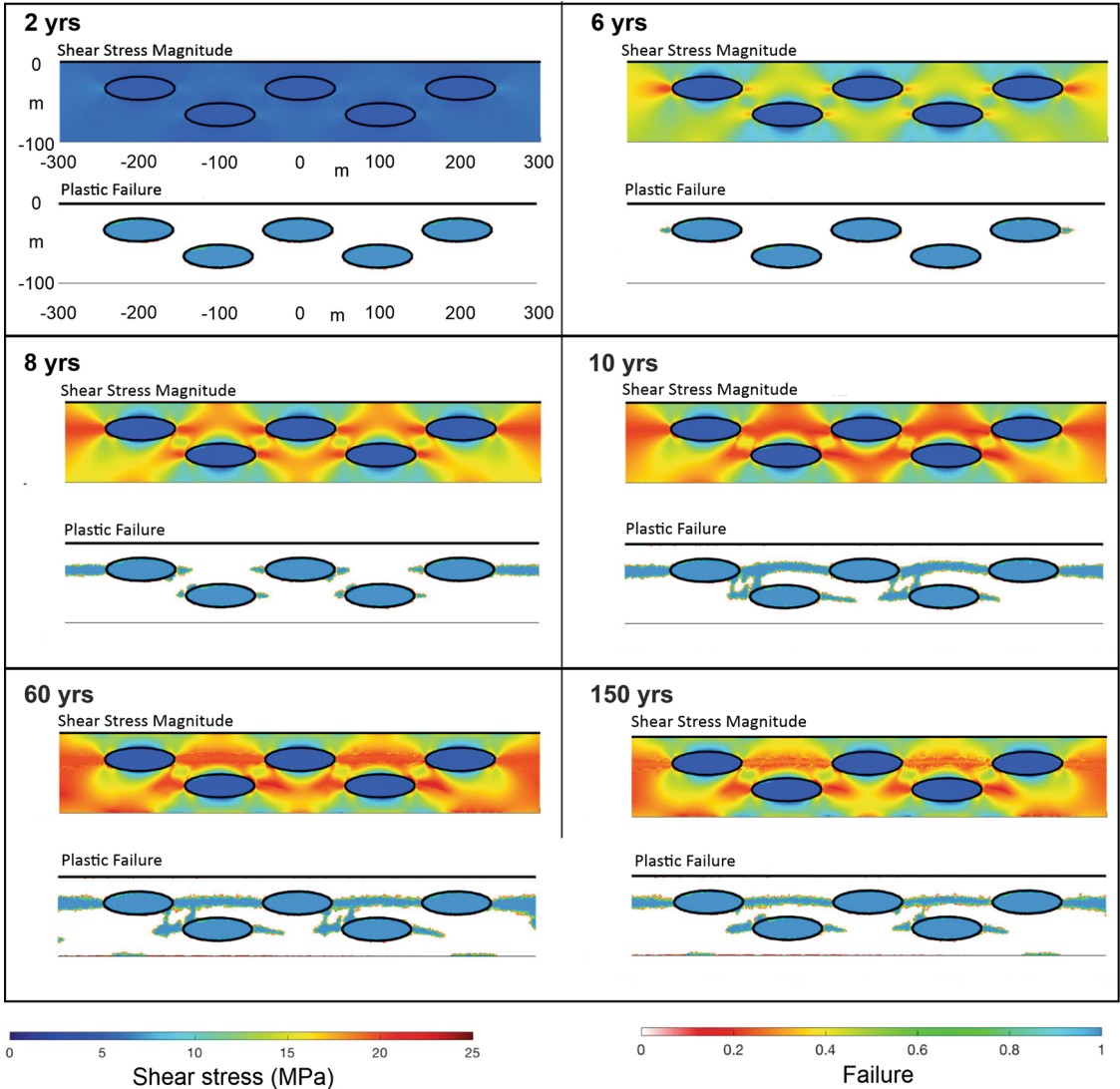

**Fig. 5 The time progression of stress and plastic failure (region failing plastically at the given moment in time) for the numerical experiment shown in Fig. 2e.** The shear channel is 100 m wide and 1 km long; each panel shows the central 600 m. The overall shear rate across the channel is assumed to be 10 mm/yr with the top moving to the right. Here the block cohesion $c = 5$ MPa is 4-times lower than the matrix cohesion ($c = 20$ MPa), the matrix viscosity is $10^{19}$ Pa s, and the block viscosity is 10-times higher ($\mu = 10^{20}$ Pa s). Note that failure of the weak blocks occurs early (after two years) while the failure of the matrix only occurs after stress concentrations have formed between the weak blocks. After 10 years both the tremorgenic blocks and the higher yield stress matrix are failing in a stable pattern. See Fig. 2 for plot conventions.

then concentrate adjacent to subducted seamounts where hydrothermal alteration is known to occur. In both the Nankai and Costa Rica forearcs this correlation appears to exist, although the basement structure beneath the sediment cover is poorly resolved, and the seismic array coverage is nonuniform (Fig. 1).

A rheological inversion could also occur as the subducting material progressively experiences higher pressure and temperature conditions that drive dehydration and metamorphism responsible for a wide range of chemical and physical reactions[54] (Fig. 6). These reactions could also drive interplays between blocks and matrix permeability and fluid production, and relative volume changes. Such physical conditions could play an important role in mélanges characterised by a clay-rich matrix where, as clay dehydrates, high pressure released fluids could preferentially migrate into the higher-bulk-permeability, higher porosity fractured blocks within the lower permeability matrix.

In general, local processes associated with rise of exotic material and fluids from a compositionally heterogeneous incoming plate can lead to strong lateral heterogeneities in both composition and mechanical strength[55]. One example can be linked to the shallow tremor patch found offshore Kii peninsula in the Nankai Trough (Fig. 1a). Seismic imaging shows a high p-wave velocity zone in the upper plate interpreted to be a plutonic body that intruded into the accreted sediments[56]. We speculate that this intrusion could have triggered local advection of hot fluids, and a subsequent strength inversion between the blocks and matrix of units dismembered by subduction channel or intraprism deformation[20,32].

Deep tremor along the megathrust is typically associated with regions where high pressure/low-temperature metamorphic transformations should take place in the subducting slab[57] (Fig. 6). Here blueschist-to-eclogite transitions and serpentinite dehydration not only increase local fluid pressures but also generate a brittle-viscous rheological contrast[23]. For example, during prograde metamorphism, heterogeneous phase transformations would reflect spatial variations in net fluid/rock ratio and/or protolith composition. Transformations of blueschists to eclogites[23,58] and/or ultramafic phases to serpentinites[59] would

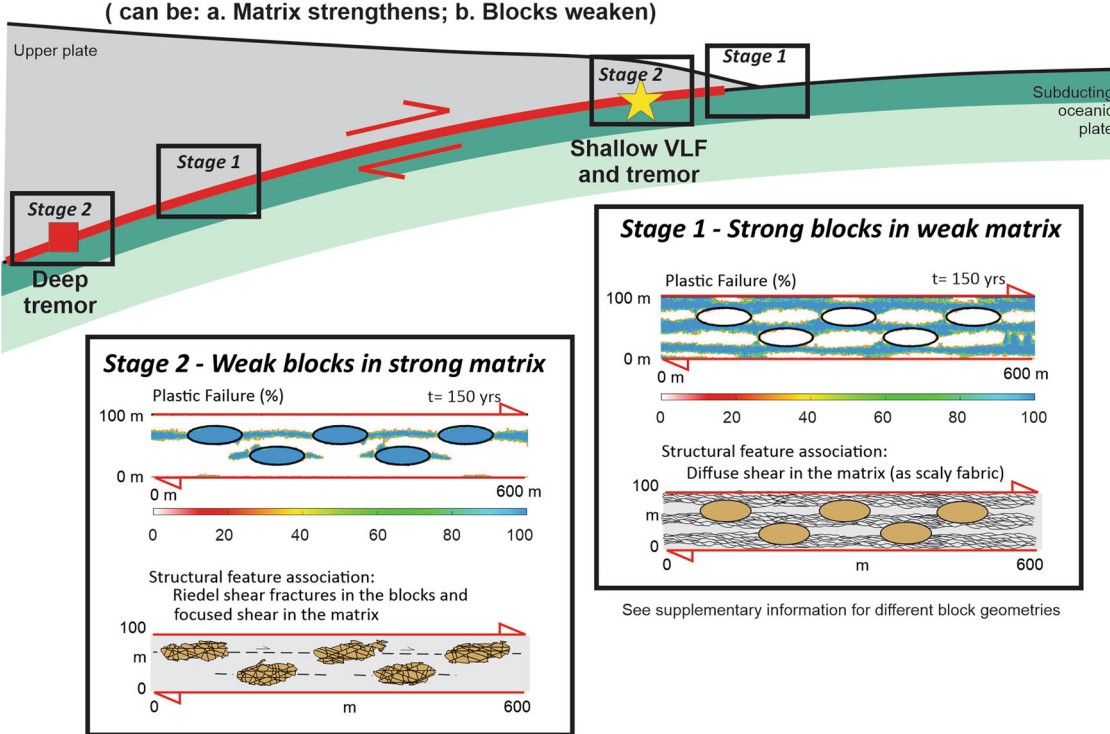

**Fig. 6 Conceptual model of potential block/matrix strength-inversion scenarios for shallow and deep non-volcanic tremor within a subduction plate boundary shear zone.** The Stage 1 and Stage 2 panels show numerical results from Fig. 2 for mechanical changes associated with this strength inversion, and geological cartoons from Fig. 3 summarising the structural characteristics linked to these numerical models. Shallow environment: Shallow Very Low Frequency (VLF) earthquakes and tremor are proposed to arise when the matrix, originally weaker than its embedded blocks, progressively becomes stronger than the blocks. Several geological processes could lead to this strengthening, including hydrothermal alteration and compaction. Deep environment: Deep non-volcanic tremor is proposed to arise because the compositionally heterogeneous blocks and matrix will undergo metamorphic transformations, in particular, the compositionally-depth-dependent blueschist-eclogite and serpentine dehydration transformations, at slightly different $p$-$T$ conditions. If the blocks undergo this fluid-releasing transition before their surrounding low-permeability matrix materials, then they could be weakened with respect to surrounding matrix both by the transition and by its associated high-pressure fluid release. See text for further discussion.

generate a strength inversion of blocks and matrix mechanically analogous to what we propose to sometimes occur along a shallow megathrust.

The relative rheology of the blocks and the matrix of subduction channel mélanges exhumed from shallow tremor source depths shows significant variations[30–32,60]. If tremorgenic conditions arise from a mechanical strength inversion in the subduction channel—i.e., weaker blocks dispersed in a stronger matrix—, we will need to reassess how stress accumulates around the zone where megathrust earthquakes nucleate. Increased tremor rates should correlate with higher strain rates within the subduction shear zone, in particular the higher strain rates associated with episodes of slow slip, and/or higher diagenetic/metamorphic fluid pressures within tremorgenic blocks, but should not imply anomalously low stresses within tremor regions (see Figs. 2d, e and 5). Instead, enhanced tremor would indicate conditions where matrix stresses could actually build. Channel stresses outside the tremorgenic blocks would remain high (Fig. 2d, e), limited only by the viscosity, strain rate, and yield strength of the stronger matrix. Large megathrust earthquakes could still easily propagate through tremorgenic zones because the more highly stressed matrix is still capable of releasing significant elastic strain energy during its rupture.

The seismological model of tremor produced by repetitive failure of asperities requires that tremorgenic failure occurs within 20–400 m-sized weak blocks, instead of stronger/quasi-rigid blocks.

The blocks have a characteristic maximum size found in seismic observations, their initial maximum size prior to breakage that eventually leads to a power-law distribution of smaller sizes. These weak blocks are embedded within a finite-thickness channel undergoing shear deformation, instead of forming seismic patches along a fault surface. This scenario needs to be integrated with quantitative friction stability models in order to learn the geological meaning of seismic asperities.

## Methods

**Numerical experiments**. The mechanical behaviour of lithospheric rocks can be described by a visco-elasto-plastic extension of the Stokes equations for creeping flow:

$$\nabla \cdot \boldsymbol{\tau} - \nabla p = -\rho \mathbf{g} \tag{1}$$

$$\frac{D\rho}{Dt} + \nabla \cdot \mathbf{u} = 0 \tag{2}$$

where $\boldsymbol{\tau}$ is deviatoric stress tensor, p is pressure, $\rho$ is density, $\mathbf{g}$ is the gravitational force, $\mathbf{u}$ is velocity field, $t$ is time, and $\nabla$ is the nabla operator. For the numerical experiment performed in this paper, Eqs. (1) and (2) are solved with LaCoDe[61], a finite elements method (FEM) code for bidimensional thermo-mechanical computations. The matrix-block aggregate behaves as a visco-elasto-plastic Maxwell body, where the total deviatoric strain rate $\dot{\boldsymbol{\varepsilon}}$ is given by the summation of the viscous, elastic, and plastic strain rate contributions:

$$\dot{\boldsymbol{\varepsilon}} = \dot{\boldsymbol{\varepsilon}}^{\text{visc}} + \dot{\boldsymbol{\varepsilon}}^{\text{el}} + \dot{\boldsymbol{\varepsilon}}^{\text{pl}} \tag{3}$$

$$\dot{\boldsymbol{\varepsilon}}^{\text{visc}} = \frac{\boldsymbol{\tau}}{2\mu} \tag{4}$$

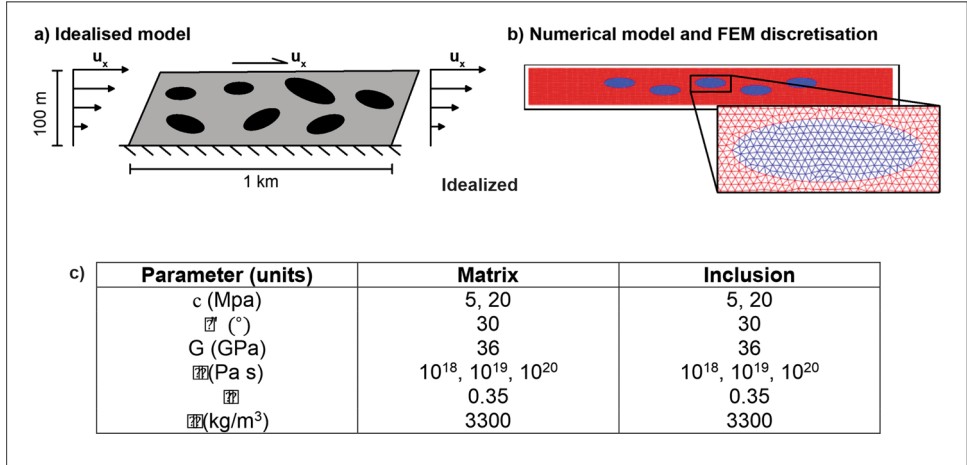

**Fig. 7 Model setup and parameters. a** Setup for an idealised model of a block-in-matrix-fabric subducting channel under simple shear boundary conditions. **b** Numerical model of the subducting channel with 5 ellipsoidal heterogeneities. Zoom-in shows the spatial discretisation of the domain using triangular elements. **c** Mechanical parameters.

$$\dot{\boldsymbol{\varepsilon}}^{\mathrm{el}} = \frac{1}{2G}\frac{D\boldsymbol{\tau}}{Dt} \qquad (5)$$

$$\dot{\boldsymbol{\varepsilon}}^{\mathrm{pl}} = \lambda\frac{\partial \mathbb{Q}}{\partial \boldsymbol{\tau}} \qquad (6)$$

where $\mu$ is viscosity, $G$ is shear modulus, $\lambda$ is the plastic multiplier and $\mathbb{Q}$ is the plastic potential. Plastic flow is computed employing a non-associative corner-free Prandtl–Reuss flow rule, and mechanical failure is defined by the pressure-sensitive Drucker–Prager yield surface $\mathscr{F}$ to approximate Mohr–Coulomb-like brittle failure:

$$\mathscr{F} \leq p\sin(\phi) + c\cos(\phi) - \tau_{II} \qquad (7)$$

where $\phi$ is the friction angle, $c$ is cohesion, and the subscript $II$ denotes the square root of the second invariant of the given tensor (i.e., $\tau_{II} = \sqrt{1/2\,\boldsymbol{\tau}:\boldsymbol{\tau}}$). If $\mathscr{F} > 0$, the deviatoric stress is corrected so that $\mathscr{F} = 0$. Thermal and chemical processes are not considered.

**Numerical model setup.** The subduction channel is numerically idealised as a two-dimensional rectangular box comprised of a two-phase aggregate under simple shear boundary conditions. Simple shear is simulated by prescribing the following boundary conditions: (i) a linearly increasing vertical profile of horizontal velocity at the side boundaries, (ii) constant horizontal velocity at the top of the domain, and (iii) zero motion at the bottom boundary. The size of the domain is $\Omega = [-500, 500] \times [-100, 0]$ m. The block-phase is represented by five identical ellipsoids, with central coordinates $x_c = [-200, -100, 0, 100, 200]$ m and $z_c = [-30, -60, -30, -60, -30]$ m, and aspect ratio of 3:1 (length: height). The semiaxes of the individual blocks are such that they represent the 10% of the area of the domain (Fig. 7a). The domain is spatially discretised by seven-node Crouziex–Raviart triangular elements in a way such that the mesh near-perfectly fits the block-matrix interface, so there are no elements crossing this interface (Fig. 7b). The mesh is constructed with the mesh generator Triangle[45].

The shear viscosity of both phases is alternated with viscosity values of $10^{18}$, $10^{19}$ and $10^{20}$ Pa s. Two-phase aggregates with strong-matrix/weak-inclusions and weak-matrix/strong-inclusions are obtained by permuting different of cohesion values, with 5 MPa for the weak phase, and 20 MPa for the strong one. All the parameters used in the numerical simulations are reported in Fig. 7c.

**Triaxial tests.** Triaxial Tests were performed at the Rock Mechanics and Physics Laboratory, British Geological Survey, UK, in an MTS 815 servo-controlled stiff frame inside a vessel capable of a confining pressure up to 140 MPa. The confining cell is fitted with external heater bands and utilises cascade control from internal and external thermocouples (accurate to ± 0.5 °C). An initial axial pre-load of 2.3 kN was applied, to ensure a stable contact and alignment of the platens. The confining pressure vessel was then closed and filled with mineral oil confining fluid. The axial pre-load was maintained whilst the confining pressure was applied at 2 MPa/min to 60 or 120 MPa; these values were chosen to approximately bracket the pressures at the up-dip limit of seismic nucleation, corresponding to 2–4 km depth[62]. At this point, whilst held in axial force and confining pressure control, the rig was heated at 2 °C/min to 60 °C to approximate the average temperature conditions at the depth of the up-dip limit of seismic nucleation[47]. Two large samples (approximately 30×30×30 cm) recovered from the Osa Mélange were cut into cylinders 54 mm in diameter and 114 mm high (Fig. 8). None of these samples

displayed orientated fabrics; they were therefore tested in a single orientation. The samples were left for approximately 1 h allowing thermal equilibrium to be reached throughout the confining fluid and the samples. Once stable, axial loading was initiated in constant axial strain rate control at a rate of $5.0 \times 10^{-6}\,\mathrm{s}^{-1}$ until macroscopic failure occurred or a significant amount of post peak-stress axial strain was recorded (between 2 and 5%). We note that one test was conducted at the higher temperature of $T = 120$ °C with a result within 2.5% of the strength at $T = 60$ °C (Fig. 8). As this is below the expected sample-to-sample variability, no further temperature studies were conducted. The axial load, axial load actuator displacement, axial stress ($\sigma_1$), differential stress ($Q = \sigma_1 - \sigma_3$), confining pressure $P_c$ ($= \sigma_2 = \sigma_3$), confining pressure actuator displacement, axial strain ($\varepsilon_{\mathrm{ax}}$), circumferential strain ($\varepsilon_{\mathrm{circ}}$) and temperature were monitored throughout at sampling frequencies of 1 s and 250 N.

At $P_c$ (confining stress) = 60 MPa and $T = 60$ °C, the volcanoclastic matrix has a strength of $\sigma_1 = 305.8$ MPa, which is about an order of magnitude higher that of the basalt blocks which is $\sigma_1 = 38.5$ MPa. The Cocos ridge basalt is nearly 3 times stronger with $\sigma_1 = 111.2$ MPa than the altered Osa Mélange basalt, but about 3 times weaker than the volcanoclastic matrix. At $P_c = 120$ MPa ($T - 60$ °C) the altered basalt is considerably stronger, as could be expected, with $\sigma_1 = 104.8$ MPa, but still remains weaker than the volcanoclastic matrix which increased in strength to $\sigma_1 = 414.8$ MPa. While the confining pressure increase has significantly reduced the strength difference between the altered basalt and the volcanoclastic matrix, the inverted rheological relationship remains. At $P_c = 120$ MPa and $T = 120$ °C, we were only able to measure the strength of the volcanoclastic matrix and observed that it did not differ significantly from its value at $T = 60$ °C: $\sigma_1 = 404.9$ MPa and $\sigma1 = 418.8$ MPa respectively. The Osa mélange basalt exhibits a multi-stage failure with an initial stress drop at 45–92% of peak stress (Fig. 3).

All sample deformation resulted in a through-going shear fracture that propagated through the sample forming an acute angle with the vertical maximum principal stress, $\sigma_1$ (all subsequent orientations are with respect to the vertical $\sigma_1$) (Fig. 8a–c). Whilst all other samples exhibit a single dominant fracture, the Osa Mélange basalt deformed at $P_c = 60$ MPa and $T = 60$ °C exhibits more complex fracturing. The deformation was accommodated by a composite fracture formed by a straight slip surface in the most altered material oriented at 20° from vertical (Fig. 8a), and an anastomosing fracture network in the less altered part of the basalt, where it partially reactivates pre-existing fractures at a range of angles from 0 to 80° from vertical (Fig. 8b). The volcanoclastic matrix deformed at $P_c = 60$ MPa and $T = 60$ °C failed by developing a single curved fracture with orientations between 24° and 41° from vertical. This is the only sample with measurable slip of ~13 mm. All the other samples developed fractures with displacements so small that they were not measurable. At $P_c = 120$ MPa and $T = 60$ °C, the Osa Mélange basalt developed a single curved fracture with orientations varying between 8° and 17° from vertical. At the same conditions, the volcanoclastic matrix failed along a curved slip surface with a maximum angle of 56° and a minimum of 8° from vertical towards to the edge of the sample; this also shows a secondary slip surface at 21° from vertical. The Cocos Ridge basalt failed by a single curved fracture with a varying orientation between 18° and 37° from vertical.

## 4. Seismological constraints on the duration-magnitude scaling of tremor sub-events.
Figure 9 shows a recent global compilation of LFE and VLF durations vs. moment release. For our study, the key finding in this data compilation is that both LFEs and VLFs have a wide variation in moment release over a small range in source durations—leading to the inference that there is a characteristic lengthscale to tremor sub-events (near-uniform source duration) but a wide variation in the

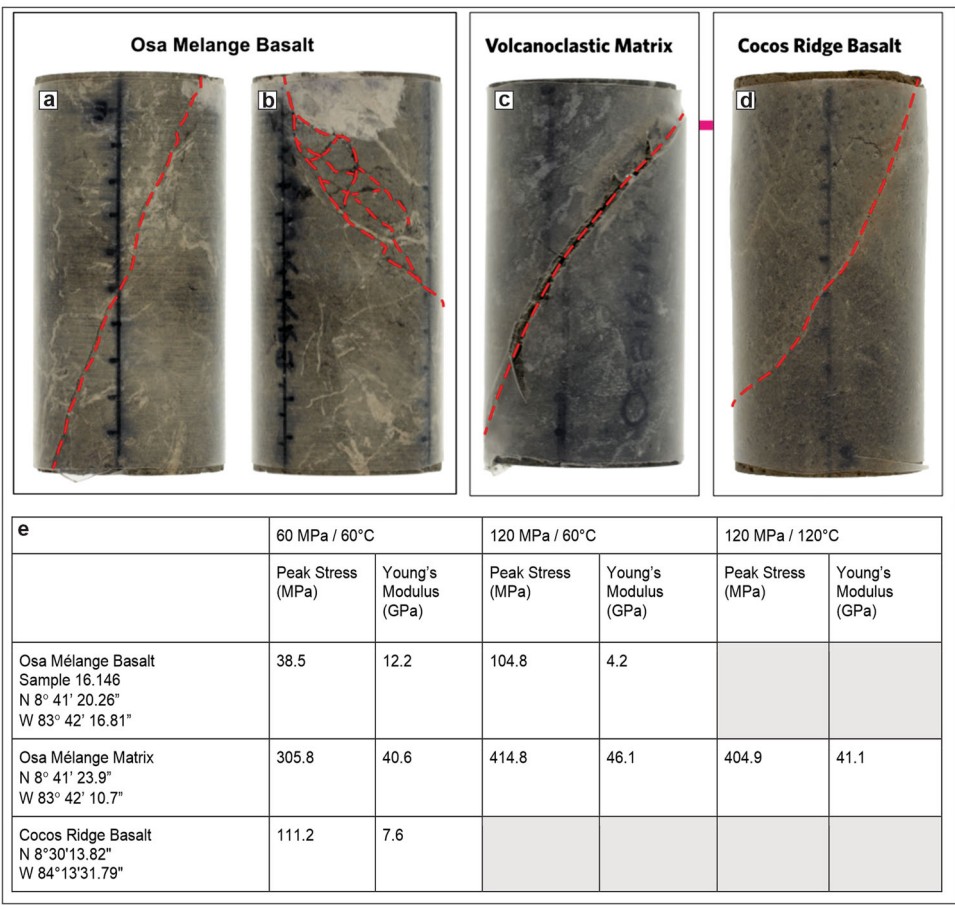

**Fig. 8 Results of Triaxial tests.** Above: Photographs of samples after experimental deformation at $P_c = 60$ MPa and $T = 60$ °C—the cylinder's dimensions are 54 × 114 mm. All samples experienced a through-going shear fracture that formed an acute angle with the vertical maximum principle stress σ1. **a** and **b** Show different deformation in the same cylinder of Osa Mélange basalt as failure is accommodated by a single tortuous fracture where no relict cores are present (A) whereas fractures exploit and reactivate fractures around and between the cores where present (**b**). **c** The volcanoclastic matrix failed by a single fracture with visible displacement and a loss of cohesion. **d** The Cocos Ridge basalt failed by a single curved fracture with visibly negligible displacement and mild tortuosity. Below: **e** Summary of mechanical results, showing peak stress and static Young's modulus values for altered basalt, volcanoclastic matrix, and Cocos Ridge basalt under 60 MPa/60 °C, 120 MPa/60 °C, and 120 MPa/120 °C.

magnitude of the (small) sub-event stress drops (wide variation in moment release). There is still an ongoing seismological debate whether LFE duration/ moment slopes are nearly flat ($M_0 \propto T^{10}$ as measured in Cascadia[7]), or instead only a little lower than the $M_0 \propto T^3$ range seen for regular earthquakes ($M_0 \propto T^{3.5\pm0.5}$ as measured in Japan[9]), or even variable between different regions. Seismological constraints on tremor stress drops are discussed in the next section.

**Seismological constraints on the characteristic stress-drop and slip-area of tremor sub-events**. It is now commonly accepted that the low-frequency earthquake (LFE) sub-events within a tremor event are associated with shear failure similar to that in regular earthquakes[5,9]. However, while the duration and magnitude (proportional to stress-drop times slip-area) of an individual LFE is relatively easy to estimate, constraining a typical LFE's stress-drop and slip-area is a much trickier seismological problem. Here we briefly review the approach and findings of a recent well-resolved seismic study which concluded that a subduction shear zone LFE's likely slip-area is of order ~20–400 m, and its stress-drop is of order ~0.03–0.4 MPa, a result broadly consistent with several previous estimates[7] (cf. [7] and references therein). The resulting clear trade-off between slip area and stress drop is also clearly illustrated in Fig. 10.

At present, the largest quiet network for the study of the seismic source properties of LFE events is the station borehole seismic network (Hi-net) in Japan. The LFE catalogue from this network has over 40,000 LFE events[9] recorded as velocity seismograms. The conventional source model assumption[63] in a seismic source analysis is that the source's power spectrum is flat at low frequencies, and has a power-law decay at frequencies above a corner frequency $f_c$. These assumptions were used to determine LFE source spectrums[9]. In this case, in principle, one can accurately measure the amplitude of the flat portion of the source spectrum which is proportional to the event's seismic moment. One can also measure the corner frequency which can be related to the size of the seismic source,

and a power-law exponent which constrains the total radiated energy of the event. These source parameters were determined by a joint parametric inversion of the spectra[9], where they solved for the joint probability density functions of each source parameter. In their study, the source parameters for an event recorded at more than one station were estimated to be the weighted means of all single-station estimates for that LFE. It was found that there was some variability in the source estimates for the same event recorded at different stations, variability that is also well known to occur in similar source estimates for normal seismic events. The major finding by Supino et al.[9] was that the seismic moment-to-corner frequency scaling exponent was of order ~−3, meaning it was similar to the moment-to-corner frequency scaling of normal seismic events, in contrast to an earlier study by Bostock et al[7]. who had determined an exponent of ~−10 for LFEs in Cascadia, but by using a different matched filter method in their analysis. This question— whether LFEs have similar scaling to normal earthquakes—is currently a major observational question in seismology. If LFEs are found to have similar scaling to regular microearthquakes and earthquakes, this would be further evidence to support the idea that LFEs involve standard shear rupture like regular seismic events.

What we wish to focus on here, however, are the typical size and stress-drop of the LFE events measured by Supino et al.[9]. Their measured range of viable size +stress-drop pairs is shown in Fig. 10. Clearly, individual constraints on either the size of the LFE slip area or its stress-drop are difficult without additional assumptions. Acceptable solutions range from ~20 to 100 m diameter faults with stress-drops of order ~1 MPa that rupture at extremely slow ~0.02β speeds (e.g., ~100 m/s speeds - Fig. 10), where β is the shear-wavespeed ($\beta = 3.7$ km/s is assumed by [9]), to 350–1600 m diameter faults with stress drops of order 10% of tidal stresses (~800–1000 Pa) that rupture at fast 0.9β speeds.

Additional observations on the triggering of tremor LFE events can be used to further narrow the range of likely tremor stress drops. In the Japan HiNet catalogue of LFE events, both solid earth tides (cf. [48]) and passing teleseismic waves (cf. [47])

| e | 60 MPa / 60°C | | 120 MPa / 60°C | | 120 MPa / 120°C | |
|---|---|---|---|---|---|---|
| | Peak Stress (MPa) | Young's Modulus (GPa) | Peak Stress (MPa) | Young's Modulus (GPa) | Peak Stress (MPa) | Young's Modulus (GPa) |
| Osa Mélange Basalt Sample 16.146 N 8° 41' 20.26" W 83° 42' 16.81" | 38.5 | 12.2 | 104.8 | 4.2 | | |
| Osa Mélange Matrix N 8° 41' 23.9" W 83° 42' 10.7" | 305.8 | 40.6 | 414.8 | 46.1 | 404.9 | 41.1 |
| Cocos Ridge Basalt N 8°30'13.82" W 84°13'31.79" | 111.2 | 7.6 | | | | |

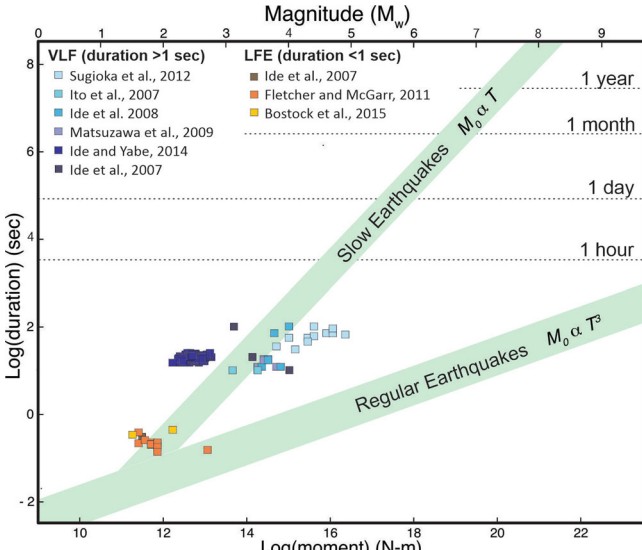

**Fig. 9 Very low frequency earthquake (VLF) and low frequency earthquake (LFE) source durations plotted against the moment release associated with these events.** Sketched are also the characteristic moment ($M_O$)-duration ($T$) trends proposed for slow ($M_O \propto T$) and regular ($M_O \propto T^3$) earthquakes. In aggregate, both VLFs and LFEs only have a small range in their duration over a two-order or larger range in the moment release of individual events. Modified from[71] and[6] which contain references to the individual data sets used in their data compilations.

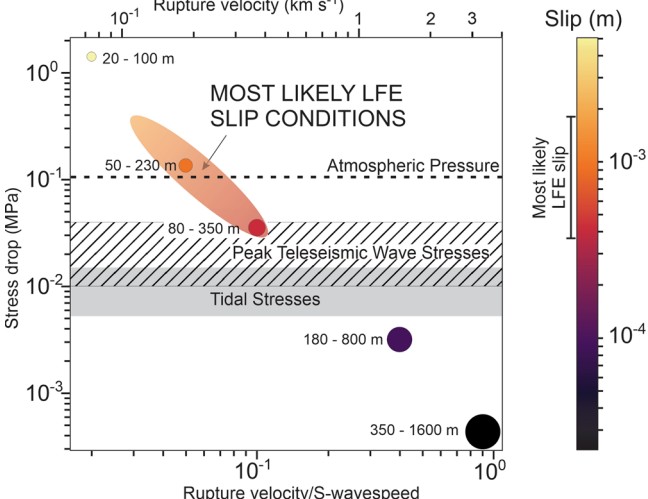

**Fig. 10 LFE stress drop, rupture dimensions, and average slip as a function of rupture velocity (modified from[9]).** The estimated constant stress drop is shown (coloured circles) for a rupture velocity that varies from 0.02 β (where β = S-wavespeed) to 0.9 β. Minimum and maximum potential rupture dimensions of the low frequency earthquakes (LFEs) are shown (circle labels), as are their average slips (circle colours). Circle sizes are scaled to the average rupture dimensions. Tidal stresses (grey box)[8] and teleseismic triggering stress levels (lined box)[65] are also shown. Reference values for the rupture velocity are given at the top of the Figure, for the assumption β = 3.7 km s⁻¹. The most likely conditions for LFE slip are highlighted by the colour-shaded regions. These involve stress drops greater than observed triggering stress perturbations due to tidal stresses or passing teleseismic waves, and rupture velocities greater than ~200 m s⁻¹, which implies LFE rupture surface diameters of order ~30–400 m, and event slips of order ~0.0004–0.002 m.

are known to trigger tremor activity. (Similar triggering of tectonic tremor is also known to occur at the well-studied Cascadia subduction shear zone and elsewhere.) Tidal stresses are on the order of 0.006–0.015 MPa[64], and the stresses induced by the passing teleseismic waves that trigger tremor are on the order of 0.01–0.04 MPa[47]. These stress magnitudes place a plausible lower-bound on the stress-drop associated with a tremor LFE event—if typical LFE stress-drops were much lower than this triggering stress level, one would anticipate that significantly smaller passing seismic wavetrains (and tides) would be linked to near-constant tremor activity, which is not observed.

This plausible lower bound of ~0.03 MPa to the stress-drop of an LFE constrains the diameter of the LFE slip patch to be ~20–400 m (Fig. 10). It further implies that the rupture speed of these events is of order 100–400 m/s (Fig. 10), considerably slower that the typical rupture speeds (1200–4000 m/s) of normal seismic events[49]. It also indicates that the stress-drop associated with an individual LFE event is of order 0.03–0.4 MPa, much lower than the ~4 MPa median stress-drop of a typical earthquake[50]. Supino et al.[9] concluded that their observations favoured that tremor signals are created by normal seismic shear rupture. Tremor shear occurs on patches of order ~20–400 m in diameter, and is associated with stress-drops of order ~0.03–0.4 MPa.

Gomberg[65] also used observations of teleseismic triggering of tremor as an argument against the hypothesis that fluid-migration induces tremor. If tremor were triggered by pressure-driven fluid migration into the LFE's slip-patch, then the observed close phase correlation between the shape of the envelope of peak teleseismic stresses and the envelope of tremor activity indicates a delay of <1–2 s between the two envelopes. In just ~1–2 s, fluids could migrate <1 m, seemingly inconsistent with the hypothesis that fluid migration into a LFE fault surface is the major trigger of tectonic tremor.

## Data availability

The data that support the findings of this study and that are not included in the Supplementary Material are available from the corresponding author. Reference to source data for figure1 are provided within the paper.

## Code availability

LaCoDe[61] is available as MATLAB source code and the geometry files used for these numerical experiments at Zenodo: https://doi.org/10.5281/zenodo.5144999.

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

## Acknowledgements
A.d.M. thanks ERC StG #758199 NEWTON for its support.

## Author contributions
P.V. conceived the research. J.M. and P.V. conceived the numerical experiments. P.V. and A.C. conducted field work. A.C. and A.O.S. conducted the triaxial experiments. A.d.M. and J.M. conducted the numerical experiments. L.A. conducted the mineralogical analyses. All authors contributed to the interpretation of the data. P.V. and J.M. wrote the paper. P.V., J.M., A.C., and A.d.M. prepared the figures.

## Competing interests
The authors declare no competing interests.
