## [Peer Review File · Nature Communications]

REVIEWER COMMENTS

Reviewer #1 (Remarks to the Author):

This study provides a new perspective on the possible generation mechanisms of tectonic tremor. The manuscript is concise and well written. The results arise from a combination of interdisciplinary elements (geological observations, laboratory measurements, numerical simulations and comparison with seismological observations), and appear to be robust. Indeed, the most interesting finding is that, differently from previous models, this study coherently accounts for the observed strong correlation between SSE and LFE activity, and for the expected failure size of LFEs.

A better understanding of how slow earthquake components are generated and interact with each other could be crucial to understanding how, and if, they could lead to the generation of large earthquakes. I suggest publication after addressing the following issues. The main question that arises from this study is how it could be possible to reconcile the proposed model of weak blocks in strong matrix with the widely observed modulation of SSEs by tidal stresses, or in other words with a kPa stress drop scale. I strongly believe that the manuscript would benefit from a broader discussion which should address this question, eventually including a possible connection with the rupture front speed ($\sim 10 \text{ km d}^{-1}$) which characterizes SSEs (e.g., Ide, 2014, 10.2183/pjab.90.259).

The Abstract (L 18-20) and Introduction (L 37-41) periods about the possible constant duration or, conversely, the self-similar behavior of VLF and LFE may be partially true, and the reported results not accurately referenced. No constant duration seems to be reported for VLFs in ref. 5-7 (or, to my knowledge, in other existing literature), while ref. 5 suggests that both VLFs and LFEs should fall into the "regular" scaling domain, id est $M_0 \propto T^3$. LFE moment-duration scaling is actually more controversial, the main question being whether their duration is almost constant (ref. 6), or it scales with the seismic moment, as observed for ordinary earthquakes (ref. 8).

Since further observations will be needed to address these questions, I would suggest presenting results about constant or nonconstant LFE (and VLF) duration as an open question to be analyzed in a further level of detail, rather than as consolidated results. Also, and more relevant to this study, a "regular" LFE momentduration scaling would imply a change in the corner frequency from $\sim 5 \text{ Hz}$ to $\sim 1 \text{ Hz}$ (for LFE seismic moment values reported in the existing literature), which corresponds to a source size ranging from $\sim 80 \text{ m}$ to $\sim 350 \text{ m}$ (ref. 8). This order of magnitude seems to be consistent with the results presented in this study.

The proposed model is able to explain how low-frequency earthquakes with a stress drop of $\sim \text{kPa}$ could enucleate close to "ordinary" earthquakes and micro-earthquakes, with a higher stress drop of several order of magnitudes. This is one of the most interesting aspect of this study, since the comparison between observed spectra of co-localized LFEs and earthquakes was a key observation to reveal the different nature of low-frequency earthquakes (Shelly et al., 2007, 10.1038/nature05666). The hypothesis of tremor triggering local ruptures of the matrix, thus "ordinary" earthquakes, could be easily verified at first order, using a well-documented catalog of earthquakes and LFEs (or tremor)? One possibility is represented by the JMA catalog, specifically looking at the tremor belt region in Japan (Shikoku-Kii-Tokai). This seems to be out of the purpose of this manuscript, still it might be interesting to discuss as a possible perspective.

The results presented in this study are in agreement with recent seismological observations related to tremor sub-events (LFEs). While the latter are well discussed in the Methods section, the connection with stress drop, slip and source size suggested by this study does not emerge strongly from the main text. I would therefore suggest to briefly expand the related discussion.

Minor comments

L 139 – 143. The explored temperature range is far from the 400 – 700 °C temperature distribution calculated for deep tectonic tremor (e.g., Ji et al., 2016, 10.1002/2016JB012912). Would it be reasonable (and technically possible) in the future to extend the experiments up to these values ?

L 265. Unit of measurement is missing.

L 350. Hi-net has more than 600 borehole stations, please correct this number (25).

Fig. 1. Please provide a title for the figure at the beginning of the caption.

Fig. 2. The figure caption is quite difficult to read and could be probably better organized. A letter should clearly indicate the beginning of a period describing each corresponding panel in the figure.

Color bars of middle- and right-panels have no label, or unit of measurement.

Fig. 4. With a log-scale for the shear stress color bar it would probably be easier to read the smaller values.

Reviewer #2 (Remarks to the Author):

I enjoyed reading this paper and particularly applaud studies that attempt to integrate field, lab, and numerical modeling. The topic is timely and should be of broad interest. I have little expertise in most aspects of the study, but hope that my comments provide guidance that will make the results more accessible to the non-expert reader. I have provided comments in the annotated manuscript returned, and below.

I suggest restructuring of the text, so that the observations are presented first, then the lab studies, and finally the numerical models. This new order might motivate the model development and validation more clearly, and provide a seemingly more objective presentation. In the Section on the weak block/strong matrix model I found it difficult to jump between the paragraphs about petrologic and lab analyses, the lab and numerical modeling. Additionally, the placement of Section 3 on the seismological constraints at the end of the paper seemed very odd, as they should motivate and provide context for the study and thus be presented first!

Addition of a more comprehensive reference list of studies of exhumed plate interface rocks would be helpful (i.e., one wonders if the Osa Melange is the only example, or one of just a few), including some that support the weak matrix/strong block model. Some of the material about the Osa Melange could be moved to the Supplement, to make room for this. Including a broader range of examples of observations would provide additional confidence in the favored model; i.e., showing the reader that the majority of observations support that model makes a much more compelling case than showing simply that a few are consistent with it.

The 'conventional' nature of the strong block/weak matrix model needs to be better documented, which could be done by providing more references. The authors show that the predictions of their numerical strong block/weak matrix model for one suite of model parameters, and that they are incompatible with some attributes of LFEs and tremor. Their rejection of this model would be strengthened by showing that this is likely a common feature of any strong block/weak matrix model; i.e., showing that the predictions broadly do not depend on parameter values assumed, block density, etc.

As noted in lines 47-51, tremor and LFEs thought to originate from asperities, or relatively strong spots embedded within a weaker matrix that slips slowly. Indeed, this is a widely adopted model, that has considerable support. Just a few studies invoking and supporting an asperity model include Ando et al. (2010, 2012), JGR; Frank et al., (2016), Sci. Adv.; Yoshida et al., (2020), and many others. Are these 'asperity' models essentially the same as the strong block/weak matrix model noted here; e.g., are 'blocks' the same thing as what is often referred to as an 'asperity'? If so, some clearly comment on why they should be rejected would be helpful. If the strong block/weak matrix model differs, the differences should be further clarified. I think the failures of these models may be noted in lines 90-91 and 107-108, but in part for

reasons noted above, these examples could be strengthened.

A broader issue related to that above, that I would have liked to see addressed, concerns the distinction between strength and friction and how both control the model of slip. As I understand it, the model proposed here does not prescribe properties that control slip or rupture speed, which determine whether slip is slow and aseismic or seismic. Traditionally slip modes are related to second order frictional properties, that aren't directly connected to strength (cohesion and viscosity), such as whether the material is velocity-strengthening or velocity-weakening, or something in between (e.g. conditionally stable, etc.). How does the model and observations proposed in this paper relate to frictional models and ideas? Can it be merged, integrated with frictional asperity models, such as the often quoted model and picture in Lay et al. (2012), JGR, and those noted above?

The explanation of 'clogging' is unclear, and the reader should not have to read reference 19 to understand what this means. Without understanding what clogging means, the contradiction noted in lines 79-81 is not apparent. Please provide a clearly explanation, for readers who are not familiar with the concept.

The discussion of shallow tremor and slow slip (lines 42-46) should acknowledge the challenges in measuring these phenomena offshore, and that their distribution is likely highly influenced by this.

The figure captions are all much too long! It is inappropriate and unnecessary to repeat what is in the text or figure legends and labels. Also, please make some of the lettering larger in the figures.

Joan Gomberg

Reviewer #3 (Remarks to the Author):

This paper uses a combination of field observations, numerical models, and rock mechanics tests to investigate the conditions in subduction zone rocks that might lead to the occurrence of low frequency earthquakes (LFEs) and slow slip events (generally referred to as slow earthquakes). The authors construct a numerical model to test how strength (viscosity and cohesion + internal friction for brittle failure) contrasts between blocks and matrix in a melange dictate the pattern of stress distribution and failure. From the models, they find that shear failure of blocks is promoted when blocks are relatively weak compared to the matrix. The field observations demonstrate fluid-assisted shear failure of basaltic blocks in a volcanoclastic matrix. Lab tests show the blocks were likely relatively weak at in situ conditions, which matches the field observations with the preferred numerical model scenarios. The authors therefore conclude that in the natural system they studied the blocks were relatively weak and that the failure of the weak blocks may have generated low frequency earthquakes. They speculate that this relative weakness of blocks is applicable to LFEs generally.

This study is of interest to the fault mechanics, structural geology, and geophysics communities it presents new results and an interesting and unusual perspective in emphasizing the importance of relatively weak blocks. The paper is generally well written, the methods are appropriate and clearly explained and the figures generally do a good job of representing the key results (I suggest some improvements to Figure 3 below). I have some included comments on the modeling and field interpretations below, which require some clarification to better support the conclusions.

My main criticism of this study is that it presents the interpretation of weak blocks as important for LFEs generally, in other words across tectonic settings and metamorphic grades. This does not seem consistent with reports in the literature that emphasize the relatively high viscosity of blocks in a relatively low viscosity matrix within subduction melanges. While the mechanical tests on block and matrix material presented here show the blocks might have been relatively weak compared to the volcanoclastic matrix in the Osa melange, I don't think the authors do a convincing job of explaining how that can be applied to other systems, for example a pelitic matrix melange (e.g. Fisher and Byrne, 1987; Kimura et al., 1991; Fagereng, 2011 etc.) or the blueschist-eclogite melange referred to in the text (Behr et al. 2018, Kotowski and Behr, 2019).

Numerous field observations in the papers listed here show boudinage of the blocks, which is interpreted to mean they had relatively high viscosity. The Osa melange therefore does not seem representative of melanges generally. The strength inversion resulting from hydrothermal cementation driven by shallow forearc pluton emplacement discussed on line 210 seems speculative and cannot apply everywhere there are slow earthquakes. Overall, I think the strength inversion idea is interesting and justified for the case study site, but I think the significance of the study findings concerning the mechanics of LFEs is overstated in the text.

Behr, W. M., Kotowski, A. J. & Ashley, K. T. Dehydration-induced rheological heterogeneity and the deep tremor source in warm subduction zones. *Geology* 46, 475-478, doi:10.1130/g40105.1 (2018).

Fagereng, Å. (2011). *Geology of the seismogenic subduction thrust interface*. Geological Society, London, Special Publications, 359(1), 55-76.

Fisher, D. & Byrne, T. STRUCTURAL EVOLUTION OF UNDERTHRUSTED SEDIMENTS, KODIAK-ISLANDS, ALASKA. *Tectonics* 6, 775-793, doi:10.1029/TC006i006p00775 (1987).

Kimura, G. & Mukai, A. UNDERPLATED UNITS IN AN ACCRETIONARY COMPLEX - MELANGE OF THE SHIMANTO BELT OF EASTERN SHIKOKU, SOUTHWEST JAPAN. *Tectonics* 10, 31-50, doi:10.1029/90tc00799 (1991).

Kotowski, A. J. & Behr, W. M. Length scales and types of heterogeneities along the deep subduction interface: Insights from exhumed rocks on Syros Island, Greece. *Geosphere* 15, 1038-1065, doi:10.1130/ges02037.1 (2019).

Modeling results

1. The geometry of the plastic failure in the matrix, and presumably in the blocks as well, seems to be highly dependent on the initial model setup with block long axes all parallel to the shear plane and centers of blocks offset sufficiently such that continuous zones of plastic failure can develop parallel to the blocks and the shear plane. This geometry is not representative of the field (e.g. Fig 3a and also numerous maps of melange elsewhere) so how much are the interpretations regarding geometry of failure regions here model-dependent? How appropriate is a simple shear boundary condition? A related point, I disagree with the statement on line 82 that blocks commonly constitute less than half of the volume of a melange. This sentence should be cited or justified better with primary data.

2. Block failure can occur at low shear stress (line 91) when the effective mean stress is low due to elevated pore pressure (as is subsequently explained on line 201). From what I can see, pore pressure is not accounted for in the numerical models. Given the field observations of hydrothermal minerals in the shear fractures cutting the blocks (Fig. 3b) and the widely held view in the literature that slow earthquakes are promoted by low effective stress conditions, the mechanical effects of elevated pore pressure seem important. How would the results of the numerical models be interpreted differently if the deformation took place under high pore fluid pressure conditions?

Field observations

The field observations on lines 116-138 are descriptive in the text, but the features described in the text are not well illustrated in figure 3. I have made suggestions for some specific labels in the figure comment below. Consider enlarging panel a to focus more on the field photo of the melange to show the lack of foliation in the matrix, absence of shear failure in the matrix, evidence for coeval matrix and block deformation, and diffuse boundaries of the blocks.

The mechanical interpretation stated on line 136, that the fault/fracture mesh pattern implies the blocks were stronger than the matrix seems incorrect to me. The blocks failed in shear, at a mean stress and differential stress large enough for shear failure, whereas the matrix failed in tension, which requires both a low mean stress and very low differential stress. Doesn't this suggest the matrix failed at lower stress conditions, i.e. was weaker?

Discussion/implications

The authors may consider broadening the discussion beginning on line 176 to include heterogeneity that involves frictional stability as well as the melange studies currently cited. As well as block viscosity, the

tendency to promote or inhibit seismic slip of blocks of different composition have been shown to be important for generation of LFEs (e.g. Phillips et al., 2020).

Phillips, N. J., Belzer, B., French, M. E., Rowe, C. D. & Ujiie, K. Frictional Strengths of Subduction Thrust Rocks in the Region of Shallow Slow Earthquakes. *J. Geophys. Res.-Solid Earth* 125, e2019JB018888, doi:<https://doi.org/10.1029/2019JB018888> (2020).

The key conclusion of this work is that “tectonic tremor is due to frequent small failure events in relatively weak blocks”. The work presented demonstrates shear failure of blocks – but what is the reason to think the failure of the blocks would have occurred at the low slip rates of low frequency earthquakes rather than slip at seismic slip rates of micro-earthquakes? The authors mention stress drop and argue that stress drop may be independent of block size. But they do not explain how the very low stress drops characteristic of LFEs may arise.

On line 197, I don’t follow the statement “In a shear zone with a down dip uniform velocity gradient, the amount of slip across the block will be a function of the block width”. Perhaps the authors could expand on this. From the relation on line 195, this might be true for a fixed stress drop – is that the reasoning?

Figures

Figure 3. Consider adding labels to panel a to help the reader decipher the key attributes of the melange. In 3b label the extensional component referred to on line 127 because it is not clear as presented. Panel c is not very useful as it is mostly black so does not illustrate anything much. Replace with a better image? Or remove? What is panel F? Are these field sketches, model outputs, conceptual models? Explanation needed in the caption and perhaps some clarification in the figure.

Figure 4. Clarify in the caption whether the plastic failure is the total plastic failure up to the time shown (finite) or whether the failure in each panel corresponds to the failure in the increment of time between panels. The latter would be a more instructive illustration.

Reviewer 1

Review of NCOMMS-20-50418

A strength inversion origin for non-volcanic tremor, by Vannucchi et al.

This study provides a new perspective on the possible generation mechanisms of tectonic tremor. The manuscript is concise and well written. The results arise from a combination of interdisciplinary elements (geological observations, laboratory measurements, numerical simulations and comparison with seismological observations), and appear to be robust. Indeed, the most interesting finding is that, differently from previous models, this study coherently accounts for the observed strong correlation between SSE and LFE activity, and for the expected failure size of LFEs.

A better understanding of how slow earthquake components are generated and interact with each other could be crucial to understanding how, and if, they could lead to the generation of large earthquakes. I suggest publication after addressing the following issues.

The main question that arises from this study is how it could be possible to reconcile the proposed model of weak blocks in strong matrix with the widely observed modulation of SSEs by tidal stresses, or in other words with a kPa stress drop scale. I strongly believe that the manuscript would benefit from a broader discussion which should address this question, eventually including a possible connection with the rupture front speed ($\sim 10 \text{ km d}^{-1}$) which characterizes SSEs (e.g., Ide, 2014, 10.2183/pjab.90.259).

The proposed mechanism for tremor still involves ‘classical’ seismic slip that will be shaped by transient higher stresses within the subduction shear channel, both the small stress perturbations associated with tidal stresses and the passage of teleseismic waves (both already discussed previously, in particular in the supplementary material where constraints from seismic observations are discussed in greater depth), and the larger potential stress perturbations associated with enhanced channel strain rates during episodes of slow slip. We have pointed this out (lines 215-219) and further modified the discussion to emphasize this point (lines 273-274). However, we don’t think it is appropriate to further enlarge the paper to discuss the interesting point the reviewer raises about a possible connection to a ‘rupture front’ propagation speed of 10 km/day that characterizes the spatial pattern of tremor migration during slow slip episodes. Note that we did previously mention, and have further emphasized that our hypothesis implies that higher tremor rates during SSEs reflect overall the higher channel shear rates during SSEs, and that a key observation to us is that tremor also occurs, at a lower rate, in between episodes of slow slip. If our hypothesis is correct, then higher tremor rates should be linked to higher channel strain rates, as is inferred from GPS measurements during SSEs. The causes for episodically higher channel strain rates could involve the mechanical properties of both blocks and matrix. We mention in the text how repeated block failure could increase matrix stresses, thereby increasing channel shear rates by enhanced dislocation creep in the matrix (again lines 215-219 and 273-274). However our proposed mechanism does not preclude other potential processes that could enhance channel strain rates and/or reduce net channel yield stresses during SSEs.

The Abstract (L 18-20) and Introduction (L 37-41) periods about the possible constant duration or, conversely, the self-similar behavior of VLF and LFE may be partially true, and the reported results not accurately referenced. No constant duration seems to be reported for VLFs in ref. 5-7 (or, to my knowledge, in other existing literature), while ref. 5 suggests that both VLFs and LFEs should fall into the “regular” scaling domain, $\text{id est } M_0 \propto T^3$.

The references do discuss the near-constant duration:

Bostock et al (2015) in their abstract say: “*LFE duration displays a weaker dependence upon moment than expected for self-similarity*”, in the main text “*A cursory examination of Figure 14 suggests that self-similarity is not honored.*” And in their concluding remarks “*The weak dependence of moment on duration is apparently at odds with the slow earthquake scaling law proposed by Ide et al. [2007] in*

which LFEs and SSEs are considered as end members. This model posits that slow earthquake moment scales with the first power of duration in contrast to regular earthquakes that scale with duration cubed as required for self-similarity.”

Gomberg et al. (2016) – ref 5 – in the main text “We note that the conventional slow M_0 proportional to T scaling is inconsistent with published VLF measurements, even considering only those from a single region. For example, Ide et al. [2008] measured M_0 proportional to $T^{3/2}$ but dismissed this deviation from the conventional scaling as an “artifact of the limited frequency range of our analysis, 0.005–0.05 Hz.” Later Ide [2008] suggested that M_0 is proportional to T^2 and proposed a different model than that in Ide et al. [2007]. Figure 1 shows published VLF measurements from Nankai, Japan, which may be explained by unbounded scaling on a fault characterized by low stress drops and slow, but still seismic, rupture propagation velocities (e.g., dotted lines (e) and (f) in Figure 1 use values reported in Matsuzawa et al. [2009]), whether considering measurements from an individual or multiple studies combined. However, despite this consistency, we do not believe that the VLF observations robustly test any scaling or physical model, as the interpretation of VLF sources also is nonunique”.

Gomberg et al. (2016) – ref. 7 – Show the following figure 8:

The reason we referenced these papers was more their figures than their words, that in any case are consistent with our summary of a “near constant” duration over two orders of magnitude. The figure we show above is clearly consistent with our description. Note also that Joan Gomberg was a reviewer of our ms and did not find our description to be inaccurate.

LFE moment-duration scaling is actually more controversial, the main question being whether their duration is almost constant (ref. 6), or it scales with the seismic moment, as observed for ordinary earthquakes (ref. 8). Since further observations will be needed to address these questions, I would suggest presenting results about constant or non-constant LFE (and VLF) duration as an open question to be analyzed in a further level of detail, rather than as consolidated results.

In our paper we only wanted to describe the current seismic observation, we do not wish to go into details of current seismic analyses approaches and debates such as the existence of a possible corner frequency related effect for LFE measurements. We feel this is better left to the seismic literature.

Also, and more relevant to this study, a “regular” LFE moment- duration scaling would imply a change in the corner frequency from ~5 Hz to ~1 Hz (for LFE seismic moment values reported in the existing literature), which corresponds to a source size ranging from ~80 m to ~350 m (ref. 8). This order of magnitude seems to be consistent with the results presented in this study.

Considering the discussion above, we added this value with the relative reference to the text (lines 46-47).

The proposed model is able to explain how low-frequency earthquakes with a stress drop of ~kPa could enucleate close to “ordinary” earthquakes and micro-earthquakes, with a higher stress drop of several order of magnitudes. This is one of the most interesting aspect of this study, since the comparison between observed spectra of co-localized LFEs and earthquakes was a key observation to reveal the different nature of low-frequency earthquakes (Shelly et al., 2007, 10.1038/nature05666). The hypothesis of tremor triggering local ruptures of the matrix, thus “ordinary” earthquakes, could be easily verified at first order, using a well-documented catalog of earthquakes and LFEs (or tremor)? One possibility is represented by the JMA catalog, specifically looking at the tremor belt region in Japan (Shikoku-Kii-Tokai). This seems to be out of the purpose of this manuscript, still it might be interesting to discuss as a possible perspective.

We thank the reviewer for the positive comment. We agree with his point of view that, although interesting, this is beyond the scope of this paper. Therefore we feel that adding a reference to this possible type of work would not strengthen the current paper.

The results presented in this study are in agreement with recent seismological observations related to tremor sub-events (LFEs). While the latter are well discussed in the Methods section, the connection with stress drop, slip and source size suggested by this study does not emerge strongly from the main text. I would therefore suggest to briefly expand the related discussion.

We added a brief discussion following the reviewer suggestion and referred to the methods for more details (lines 228-236).

Minor comments

L 139 – 143. The explored temperature range is far from the 400 – 700 °C temperature distribution calculated for deep tectonic tremor (e.g., Ji et al., 2016, 10.1002/2016JB012912). Would it be reasonable (and technically possible) in the future to extend the experiments up to these values ?

Yes, but this would require a different apparatus.

L 265. Unit of measurement is missing.

Thank you for pointing this out. Now added.

L 350. Hi-net has more than 600 borehole stations, please correct this number (25).

Thank you, now corrected.

Fig. 1. Please provide a title for the figure at the beginning of the caption.

We added a title.

Fig. 2. The figure caption is quite difficult to read and could be probably better organized. A letter should clearly indicate the beginning of a period describing each corresponding panel in the figure.

Color bars of middle- and right-panels have no label, or unit of measurement.

We changed the figure caption and made it easier to read following the reviewer's suggestions. Text in figure 2 have been enlarged. Strain is dimensionless, but we added % to the failure panels to make it clearer that this was also dimensionless.

Fig. 4. With a log-scale for the shear stress color bar it would probably be easier to read the smaller values.

We enlarged the whole figure.

Reviewer #2 (Remarks to the Author):

I enjoyed reading this paper and particularly applaud studies that attempt to integrate field, lab, and numerical modeling. The topic is timely and should be of broad interest. I have little expertise in most aspects of the study, but hope that my comments provide guidance that will make the results more accessible to the non-expert reader. I have provided comments in the annotated manuscript returned, and below.

I suggest restructuring of the text, so that the observations are presented first, then the lab studies, and finally the numerical models. This new order might motivate the model development and validation more clearly, and provide a seemingly more objective presentation. In the Section on the weak block/strong matrix model I found it difficult to jump between the paragraphs about petrologic and lab analyses, the lab and numerical modeling.

We tried to follow the reviewer's proposed structure. We have an introduction followed by the state of the art, then we introduce field observations that suggest an alternative scenario. These observations are followed by the lab measurements that support the scenario. And finished by numerical models that also support and further analyze the scenario.

Additionally, the placement of Section 3 on the seismological constraints at the end of the paper seemed very odd, as they should motivate and provide context for the study and thus be presented first!

In response of this comment and reviewer #1 comment we now have discussed the seismic constraints earlier in the main text of the paper.

Addition of a more comprehensive reference list of studies of exhumed plate interface rocks would be helpful (i.e., one wonders if the Osa Melange is the only example, or one of just a few), including some that support the weak matrix/strong block model. Some of the material about the Osa Melange could be moved to the Supplement, to make room for this. Including a broader range of examples of observations would provide additional confidence in the favored model; i.e., showing the reader that the majority of observations support that model makes a much more compelling case than showing simply that a few are consistent with it.

There are no published studies, so far, that combine field, lab testing and modelling. The examples of plate boundary shear zones studied by structural geologists around the world are interpreted following the conventional model (and we reference them in the paper) and we show that those are not tectogenic. Now that we have a new reference we are indeed studying a new example in Italy that shows the same characteristics involving serpentine and sediments.

The 'conventional' nature of the strong block/weak matrix model needs to be better documented, which could be done by providing more references. The authors show that the predictions of their

numerical strong block/weak matrix model for one suite of model parameters, and that they are incompatible with some attributes of LFEs and tremor. Their rejection of this model would be strengthened by showing that this is likely a common feature of any strong block/weak matrix model; i.e., showing that the predictions broadly do not depend on parameter values assumed, block density, etc.

More references have been added to our initial presentation of the strong block-in-weak matrix scenario.

Our analysis of the problems with the strong blocks in weak matrix model was based on both analytical reasoning and a suite of three different numerical experiments (fig. 2 A,B,C) which were 50% of the numerical model that we show. In the revision 8 more models have been added and discussed in the supplement that discuss somewhat less idealized and 'more geological' examples spanning a wider range of parameter values and blob geometries. While these examples all do show similar behavior to the case studies discussed in the text — thereby addressing the reviewer's concern — we feel there is no space (or figure space) in the main text of this short paper to include this more in depth discussion. Instead, we feel this issue would be best explored as the main topic of a future long-format publication. In fact, we hope to soon supervise a future research project that will explore this issue in greater depth.

As noted in lines 47-51, tremor and LFEs thought to originate from asperities, or relatively strong spots embedded within a weaker matrix that slips slowly. Indeed, this is a widely adopted model, that has considerable support. Just a few studies invoking and supporting an asperity model include Ando et al. (2010, 2012), JGR; Frank et al., (2016), Sci. Adv.; Yoshida et al., (2020), and many others. Are these 'asperity' models essentially the same as the strong block/weak matrix model noted here; e.g., are 'blocks' the same thing as what is often referred to as an 'asperity'? If so, some clearly comment on why they should be rejected would be helpful. If the strong block/weak matrix model differs, the differences should be further clarified. I think the failures of these models may be noted in lines 90-91 and 107-108, but in part for reasons noted above, these examples could be strengthened.

We agree and we added a sentence to the end of the "Strong blocks in weak matrix" paragraph to further clarify that our analysis implies that strong blocks cannot be a geological proxy for tremorigenic asperities (Lines 130-131).

A broader issue related to that above, that I would have liked to see addressed, concerns the distinction between strength and friction and how both control the model of slip. As I understand it, the model proposed here does not prescribe properties that control slip or rupture speed, which determine whether slip is slow and aseismic or seismic. Traditionally slip modes are related to second order frictional properties, that aren't directly connected to strength (cohesion and viscosity), such as whether the material is velocity-strengthening or velocity-weakening, or something in between (e.g. conditionally stable, etc.). How does the model and observations proposed in this paper relate to frictional models and ideas? Can it be merged, integrated with frictional asperity models, such as the often quoted model and picture in Lay et al. (2012), JGR, and those noted above?

Our model is implying that tremor is a frictional failure process. Regarding the velocity-weakening or velocity-strengthening slip modes, we hope to do this in future work. We feel that this discussion, while very important, is outside of the main scope of this short paper.

The explanation of 'clogging' is unclear, and the reader should not have to read reference 19 to understand what this means. Without understanding what clogging means, the contradiction noted in lines 79-81 is not apparent. Please provide a clearly explanation, for readers who are not familiar with the concept.

We added a definition of clogging to the text where we introduce the term. (Line 89-91)

The discussion of shallow tremor and slow slip (lines 42-46) should acknowledge the challenges in

measuring these phenomena offshore, and that their distribution is likely highly influenced by this.

We modified the sentence to make it clear that the distribution depends on the spatial coverage of the measurements.

The figure captions are all much too long! It is inappropriate and unnecessary to repeat what is in the text or figure legends and labels. Also, please make some of the lettering larger in the figures.

We have modified caption and figures.

Joan Gomberg

Next are the comments and responses to Reviewer #2's comments on their annotated manuscript that they also included – line # refers to the current version.

Line 31 Unclear where this number comes from, but slip speeds are orders of magnitude smaller and rupture speeds (more relevant) can be orders of magnitude larger? Please clarify

We clarify the sentence and we added a recent review reference for slip rates

Line 48-56. The discussion of shallow tremor and slow slip (lines 42-46) should acknowledge the challenges in measuring these phenomena offshore, and that their distribution is likely highly influenced by this.

We modified the sentence to make it clear that depended on the spatial coverage of the measurements.

Line 56 This is likely just a reflection of the fact that tremor is more easily detected than aseismic slip, rather than tremor happening in the absence of aseismic slip (at least one can't rule this out, and evidence is mounting that aseismic slip happens more often than is typically detected with just standard GPS processing). That this could be a detection difference rather than a real physical one should be noted.

We did not change the sentence because we just reported the current state of observations.

Line 57-61. As noted in lines 47-51, tremor and LFEs thought to originate from asperities, or relatively strong spots embedded within a weaker matrix that slips slowly. Indeed, this is a widely adopted model, that has considerable support. Just a few studies invoking and supporting an asperity model include Ando et al. (2010, 2012), JGR; Frank et al., (2016), Sci. Adv.; Yoshida et al., (2020), and many others. Are these 'asperity' models essentially the same as the strong block/weak matrix model noted here; e.g., are 'blocks' the same thing as what is often referred to as an 'asperity'? If so, some clearly comment on why they should be rejected would be helpful. If the strong block/weak matrix model differs, the differences should be further clarified. I think the failures of these models may be noted in lines 90-91 and 107-108, but in part for reasons noted above, these examples could be strengthened

We have added several sentences at the very end of the paper to address this important conceptual comment

Line 110. If the cohesion in the matrix is higher than the blocks, doesn't this make the matrix stronger - at least in one sense? Maybe this number is mixed up (the figure indicates it may be)?

Thank you for noticing this glitch in the text. We modified the sentence – the figure and figure caption were correct.

Line 118-119. Unless I don't understand, the blocks have lower cohesion than the matrix (5 MPa vs 20 MPa for the latter)?

Clarified in the text.

Line 215. The term 'microseismic' is confusing and ambiguous (what is 'micro'?) - what type of event does this refer to?

We changed microseismic to seismic.

Line 219-220 This linkage between tremor/slow slip rates and earthquake rates is only observed in a few places. If this model is broadly applicable, why isn't such a linkage more commonly observed?

The fact that it is observed is what is important here. How generally it is observed is a question for future seismological research that is beyond the scope of this paper.

Line 221. Why just 'micro-earthquakes', and not earthquakes generally?

We use micro-earthquakes because we want to stress that even very small earthquakes have this property.

Line 244. While plausibly related to tremor generation, subducted seamounts seemingly must only account for tiny fractions of subduction interfaces and regions where tremor exists. Is there a more general, or widely distributed feature or mechanism for creating this strength inversion?

We do mention other possible mechanisms in the following two paragraphs. We agree that this is an important area for future exploration

Line 255. As for the seamounts, this explanation only explains tremor observations in a few places? Again, is there a more broadly applicable explanation?

We can imagine many possible ways to envision strength inversion related to diagenesis/metamorphism, but we mentioned three possible examples: "ocean floor" hydrothermalism, "magmatic intrusion-related" hydrothermalism and progressive eclogitization.

Line 402-415. This degree of detail seems out of place here; perhaps shorten this text?

We feel that non-seismologists will appreciate this level of detail.

Line 476-480. Replace with something like "showing the distribution of slip modes and related tectonic structures".

For this caption we prefer to keep this level of detail since this figure is a compilation of different datasets so that we can give appropriate credit as well as making it easier to the reader to follow the figure.

Line 484-486. This belongs in the text, not the caption

We moved the sentence about distribution to the text.

Line 490. This caption is WAY too long. Do not repeat what is in the text or noted in the figure!

We modified and shortened the caption by 25%.

Line 510-513. Shorten.

Done

Line 525-528

This is a reasonable caption! The print in the figures needs to be enlarged.

Done.

Reviewer #3 (Remarks to the Author):

This paper uses a combination of field observations, numerical models, and rock mechanics tests to investigate the conditions in subduction zone rocks that might lead to the occurrence of low frequency earthquakes (LFEs) and slow slip events (generally referred to as slow earthquakes). The authors construct a numerical model to test how strength (viscosity and cohesion + internal friction for brittle failure) contrasts between blocks and matrix in a melange dictate the pattern of stress distribution and failure. From the models, they find that shear failure of blocks is promoted when blocks are relatively weak compared to the matrix. The field observations demonstrate fluid-assisted shear failure of basaltic blocks in a volcanoclastic matrix. Lab tests show the blocks were likely relatively weak at in situ conditions, which matches the field observations with the preferred numerical model scenarios. The authors therefore conclude that in the natural system they studied the blocks were relatively weak and that the failure of the weak blocks may have generated low frequency earthquakes. They speculate that this relative weakness of blocks is applicable to LFEs generally.

This study is of interest to the fault mechanics, structural geology, and geophysics communities it presents new results and an interesting and unusual perspective in emphasizing the importance of relatively weak blocks. The paper is generally well written, the methods are appropriate and clearly explained and the figures generally do a good job of representing the key results (I suggest some improvements to Figure 3 below). I have some included comments on the modeling and field interpretations below, which require some clarification to better support the conclusions.

My main criticism of this study is that it presents the interpretation of weak blocks as important for LFEs generally, in other words across tectonic settings and metamorphic grades. This does not seem consistent with reports in the literature that emphasize the relatively high viscosity of blocks in a relatively low viscosity matrix within subduction melanges. While the mechanical tests on block and matrix material presented here show the blocks might have been relatively weak compared to the volcanoclastic matrix in the Osa melange, I don't think the authors do a convincing job of explaining how that can be applied to other systems, for example a pelitic matrix melange (e.g. Fisher and Byrne, 1987; Kimura et al., 1991; Fagereng, 2011 etc.) or the blueschist-eclogite melange referred to in the text (Behr et al. 2018, Kotowski and Behr, 2019). Numerous field observations in the papers listed here show boudinage of the blocks, which is interpreted to mean they had relatively high viscosity. The Osa melange therefore does not seem representative of melanges generally.

“Boudinage of the blocks, which is interpreted to mean they had relatively high viscosity” – In our experiment the basalt blocks breaks more readily than the matrix – therefore they are more competent or stiffer (i.e. more viscous) according with the reviewer's statement (see Figure 2D and E in particular where the blocks are failing). In terms of strength, though, and their ability to fail, the

blocks **are weaker**, because the matrix does not break, therefore is tougher. We have further modified the text (lines 195-196 and 206-208) to make this point clear – this is why we have table 1 in the supplement.

The strength inversion resulting from hydrothermal cementation driven by shallow forearc pluton emplacement discussed on line 210 seems speculative and cannot apply everywhere there are slow earthquakes. Overall, I think the strength inversion idea is interesting and justified for the case study site, but I think the significance of the study findings concerning the mechanics of LFEs is overstated in the text.

We have expanded the text (Lines 254-259) and we have made clear that this is an example of a possible local reason to have injection of exotic fluids in the subduction plate shear zone.

Behr, W. M., Kotowski, A. J. & Ashley, K. T. Dehydration-induced rheological heterogeneity and the deep tremor source in warm subduction zones. *Geology* 46, 475-478, doi:10.1130/g40105.1 (2018).
Fagereng, Å. (2011). *Geology of the seismogenic subduction thrust interface*. Geological Society, London, Special Publications, 359(1), 55-76.
Fisher, D. & Byrne, T. STRUCTURAL EVOLUTION OF UNDERTHRUSTED SEDIMENTS, KODIAK-ISLANDS, ALASKA. *Tectonics* 6, 775-793, doi:10.1029/TC006i006p00775 (1987).
Kimura, G. & Mukai, A. UNDERPLATED UNITS IN AN ACCRETIONARY COMPLEX - MELANGE OF THE SHIMANTO BELT OF EASTERN SHIKOKU, SOUTHWEST JAPAN. *Tectonics* 10, 31-50, doi:10.1029/90tc00799 (1991).
Kotowski, A. J. & Behr, W. M. Length scales and types of heterogeneities along the deep subduction interface: Insights from exhumed rocks on Syros Island, Greece. *Geosphere* 15, 1038-1065, doi:10.1130/ges02037.1 (2019).

Modeling results

1. The geometry of the plastic failure in the matrix, and presumably in the blocks as well, seems to be highly dependent on the initial model setup with block long axes all parallel to the shear plane and centers of blocks offset sufficiently such that continuous zones of plastic failure can develop parallel to the blocks and the shear plane. This geometry is not representative of the field (e.g. Fig 3a and also numerous maps of melange elsewhere) so how much are the interpretations regarding geometry of failure regions here model-dependent? How appropriate is a simple shear boundary condition? A related point, I disagree with the statement on line 82 that blocks commonly constitute less than half of the volume of a melange. This sentence should be cited or justified better with primary data.

We have removed this sentence in the revised text. We continue to focus on the idealized examples in the main text because they most clearly illustrate the mechanical behavior that we wish to explore — there is always a tension between making a model simple enough to clearly understand its behaviour, and making it ‘too simplified’. To address the reviewer’s concern, we also now include a more in-depth discussion of a wider range of models in the supplement, including a case based on an observed block distribution in the field. All of our additional examples show the key behavioural aspects that we highlight in the main text and figures — the basic contrast between strong block/weak matrix and weak block/strong matrix behavior. The new supplementary discussion includes a brief discussion of an artificial aspect induced by the simple shear boundary condition that has strongly influenced the channel model results of Beall et al. (2019) — we agree with the reviewer that the choice of appropriate BCs, including even the mechanical definition of the channel itself, is an important issue in modelling-based exploration, numerical as well as lab-based, that needs to be better explored in future work.

2. Block failure can occur at low shear stress (line 91) when the effective mean stress is low due to elevated pore pressure (as is subsequently explained on line 201). From what I can see, pore pressure is not accounted for in the numerical models. Given the field observations of hydrothermal minerals in

the shear fractures cutting the blocks (Fig. 3b) and the widely held view in the literature that slow earthquakes are promoted by low effective stress conditions, the mechanical effects of elevated pore pressure seem important. How would the results of the numerical models be interpreted differently if the deformation took place under high pore fluid pressure conditions?

Fluids in the blocks can only increase the ease of the blocks to break and therefore increase their weakness. In the experiments a high block fluid pressure would be equivalent to further reducing the effective cohesion. We now explicitly put this point in our discussion in the text (lines 195-196,196-198).

Field observations

The field observations on lines 116-138 are descriptive in the text, but the features described in the text are not well illustrated in figure 3. I have made suggestions for some specific labels in the figure comment below. Consider enlarging panel a to focus more on the field photo of the melange to show the lack of foliation in the matrix, absence of shear failure in the matrix, evidence for coeval matrix and block deformation, and diffuse boundaries of the blocks.

We enlarged the figure.

The mechanical interpretation stated on line 136, that the fault/fracture mesh pattern implies the blocks were stronger than the matrix seems incorrect to me. The blocks failed in shear, at a mean stress and differential stress large enough for shear failure, whereas the matrix failed in tension, which requires both a low mean stress and very low differential stress. Doesn't this suggest the matrix failed at lower stress conditions, i.e. was weaker?

The reviewer possibly made a typo in the comment about our interpretation of the blocks as "stronger". We believe the reviewer meant "weaker", which is what we had said. Aside from this, the reviewer comment would be correct if the blocks and the matrix had the same tensile strength and cohesion, but we think that this mélange (as most melanges) is truly heterogeneous, therefore the matrix and the blocks have different material properties. The matrix, for example, would have a higher cohesion which would increase its resistance to shear failure. The result would be that the Griffith/Coulomb failure envelope of the matrix would be "higher" than the failure envelope of the blocks and therefore for the same differential stress the blocks would fail in shear while the matrix would fail in tension. As a corollary, the matrix also needs to develop a high fluid pressure to fail in tension (a fluid pressure that the blocks could not sustain without prior shear failure). To make this clearer we added a reference of material properties to the sentence indicated by the reviewer.

Discussion/implications

The authors may consider broadening the discussion beginning on line 176 to include heterogeneity that involves frictional stability as well as the melange studies currently cited. As well as block viscosity, the tendency to promote or inhibit seismic slip of blocks of different composition have been shown to be important for generation of LFEs (e.g. Phillips et al., 2020).

Phillips, N. J., Belzer, B., French, M. E., Rowe, C. D. & Ujiie, K. Frictional Strengths of Subduction Thrust Rocks in the Region of Shallow Slow Earthquakes. *J. Geophys. Res.-Solid Earth* 125, e2019JB018888, doi:<https://doi.org/10.1029/2019JB018888> (2020).

We thank the reviewer for this comment, and enjoyed reading this paper when we saw it. We tried hard to find a way to insert a few sentences of discussion of this paper into our revised text, but were unable to do so. There were two difficulties in fitting this suggested discussion into our short manuscript: (1) we do not discuss velocity-strengthening or velocity-weakening behaviour in our

laboratory experiments because they were designed in a different way than the lab experiments presented in the Phillips et al. paper. (2) in our discussion section ‘Implications for subduction zone tremor’, we found no way to add a sentence or two to mention this point without also adding a full paragraph of background to provide context for our sentences. Therefore, we think this discussion would be much better suited to be included in a longer paper in the future.

The key conclusion of this work is that “tectonic tremor is due to frequent small failure events in relatively weak blocks”. The work presented demonstrates shear failure of blocks – but what is the reason to think the failure of the blocks would have occurred at the low slip rates of low frequency earthquakes rather than slip at seismic slip rates of micro-earthquakes? The authors mention stress drop and argue that stress drop may be independent of block size. But they do not explain how the very low stress drops characteristic of LFEs may arise.

We have discussed this point in the method section on seismic observations. We propose that the low stress drops reflects the low cohesion of the fractured block material. The idea is that blocks have anomalously weak failure surfaces that have very low stress drops when they fail. We have added this point to the discussion (lines 229-237)

On line 197, I don’t follow the statement “In a shear zone with a down dip uniform velocity gradient, the amount of slip across the block will be a function of the block width”. Perhaps the authors could expand on this. From the relation on line 195, this might be true for a fixed stress drop – is that the reasoning?

We have added the word “thickness” to that description of width to try to make this geometrical observation clearer.

Figures

Figure 3.

Consider adding labels to panel a to help the reader decipher the key attributes of the melange.

Added labels

In 3b label the extensional component referred to on line 127 because it is not clear as presented.

Added arrows

Panel c is not very useful as it is mostly black so does not illustrate anything much. Replace with a better image? Or remove?

Made it brighter

What is panel F? Are these field sketches, model outputs, conceptual models? Explanation needed in the caption and perhaps some clarification in the figure.

Done

Figure 4. Clarify in the caption whether the plastic failure is the total plastic failure up to the time shown (finite) or whether the failure in each panel corresponds to the failure in the increment of time between panels. The latter would be a more instructive illustration.

In the revised figure caption, we clarify that plastic failure refer to the region failing plastically at that given moment in time.

REVIEWER COMMENTS

Reviewer #1 (Remarks to the Author):

The response from the authors is detailed, and fully answers to almost all comments. I have a few additional small comments.

Black text: This review

Blue text: First review

Red text: Authors' response

General comment: in the authors' response, some references to the manuscript lines seem to be not correct, probably shifted.

L 19-21, 38-40.

The Abstract (L 18-20) and Introduction (L 37-41) periods about the possible constant duration or, conversely, the self-similar behavior of VLF and LFE may be partially true, and the reported results not accurately referenced. No constant duration seems to be reported for VLFs in ref. 5-7 (or, to my knowledge, in other existing literature), while ref. 5 suggests that both VLFs and LFEs should fall into the "regular" scaling domain, id est $M_0 \propto T^3$.

The references do discuss the near-constant duration:

Bostock et al (2015) in their abstract say: "LFE duration displays a weaker dependence upon moment than expected for self-similarity", in the main text "A cursory examination of Figure 14 suggests that self-similarity is not honored." And in their concluding remarks "The weak dependence of moment on duration is apparently at odds with the slow earthquake scaling law proposed by Ide et al. [2007] in which LFEs and SSEs are considered as end members. This model posits that slow earthquake moment scales with the first power of duration in contrast to regular earthquakes that scale with duration cubed as required for self-similarity."

Gomberg et al. (2016) – ref 5 – in the main text "We note that the conventional slowM0 proportional to T scaling is inconsistent with published VLF measurements, even considering only those from a single region. For example, Ide et al. [2008] measured M0 proportional to $T^{3/2}$ but dismissed this deviation from the conventional scaling as an "artifact of the limited frequency range of our analysis, 0.005–0.05 Hz." Later Ide [2008] suggested that M0 is proportional to T^2 and proposed a different model than that in Ide et al. [2007]. Figure 1 shows published VLF measurements from Nankai, Japan, which may be explained by unbounded scaling on a fault characterized by low stress drops and slow, but still seismic, rupture propagation velocities (e.g., dotted lines (e) and (f) in Figure 1 use values reported in Matsuzawa et al. [2009]), whether considering measurements from an individual or multiple studies combined. However, despite this consistency, we do not believe that the VLF observations robustly test any scaling or physical model, as the interpretation of VLF sources also is nonunique".

Gomberg et al. (2016) – ref. 7 – Show the following figure 8:

The reason we referenced these papers was more their figures than their words, that in any case are consistent with our summary of a "near constant" duration over two orders of magnitude. The figure we show above is clearly consistent with our description. Note also that Joan Gomberg was a reviewer of our ms and did not find our description to be inaccurate.

I am sorry but I have to confirm my comment, I will try to better detail. As you wrote, Bostock et al. (2015) showed a near-constant duration ($T \propto M_0^{1/10}$, very different from the regular scaling $T \propto M_0^{1/3}$) for LFEs in Cascadia. But this result is reported for LFEs, not VLFs.

A near-constant duration for VLFs should mean a weak duration dependence upon moment, with a very small exponent (similarly to Bostock et al. 2015). In ref. 5-7 (now 6-8), I am not able to find a similar result. On the contrary (and similar to what you wrote in your answer!), the last lines of the abstract of ref. 8 seem to be a very good picture of the current literature about VLF scaling: "Given

the non-uniqueness in possible source durations, we suggest it is premature to draw conclusions about VLF event sources or how they scale”.

L 19-21, 38-40.

LFE moment-duration scaling is actually more controversial, the main question being whether their duration is almost constant (ref. 6), or it scales with the seismic moment, as observed for ordinary earthquakes (ref. 8). Since further observations will be needed to address these questions, I would suggest presenting results about constant or non- constant LFE (and VLF) duration as an open question to be analyzed in a further level of detail, rather than as consolidated results.

In our paper we only wanted to describe the current seismic observation, we do not wish to go into details of current seismic analyses approaches and debates such as the existence of a possible corner frequency related effect for LFE measurements. We feel this is better left to the seismic literature.

I do agree with just describing the current seismic observations. However, seismic observations of Supino et al. (2020) show a self-similar moment-duration scaling for LFEs in Nankai, which is the opposite of what you described. The simplest way to describe the current seismic observation could be that the LFE moment-duration scaling changes depending on the subduction zone analyzed.

You may choose not to consider the results of Supino et al. (2020). However, the results of Supino et al. (2020) are quite used in this manuscript.

L 40-41. Is Figure 1 showing information about the near-constant <1s duration of LFEs ?

Reviewer #2 (Remarks to the Author):

The authors have done a fine job addressing the Reviewers' concerns. I think this paper is ready for publication.

Reviewer #3 (Remarks to the Author):

Review of 'A strength inversion origin for non-volcanic tremor' by Vannucchi et al.

This is the second time I have reviewed this manuscript, previously as reviewer 3. In general, the authors have responded constructively to the previous round of comments from reviewers.

This study integrates numerical modeling and laboratory rock mechanics measurements to demonstrate that the fracture patterns within blocks in the Osa Melange, Costa Rica, formed under conditions where the blocks were relatively weak compared to the surrounding matrix.

The manuscript has two major problems. First, it sets out to disprove the "strong blocks" model. However, the author's arguments about the mechanical considerations and geological predictions of the strong block model are inconsistent and without observational basis. The paper would be stronger if it acknowledged the strong block model is a valid hypothesis. Second, the manuscript attempts to portray the "weak block" model as applicable to LFEs generally. While the study results do a good job of explaining the Osa Melange features, I disagree that the weak block model can be generally representative. The volcanoclastic matrix of the Osa Melange is atypical. Subduction melanges exhumed from shallow depth more commonly contain a pelitic matrix. The fractures observed in the blocks of the Osa melange are not seen in other melanges (e.g. see the excellent map of the Mugi Melange in Kimura et al. (2012)). The key parameter that changes between the strong block/weak matrix and weak block/strong matrix is the cohesion of the two components. The cohesion of pelitic rocks is typically of the order of 1 MPa or less (e.g. see compilations in the text books by Zoback (Reservoir Geomechanics), Jaeger, Cook and Zimmerman (Fundamentals of Rock Mechanics)), so would presumably always be less than a lithified rock.

Overall, I think the strength inversion idea is interesting and justified for the case study site, but I think the significance of the study findings concerning the mechanics of LFEs is overstated in the text. Furthermore, none of the seismological characteristics of LFEs discussed in the manuscript seem to imply the weak block model is relevant or preferred. I have highlighted below some areas where the work is incorrect or unsubstantiated. I recommend the authors modify the text to rectify the problems outlined above. I think this could be done with minor revisions to the text, although these revisions do need to address the core theme of the paper.

Kimura, G., Yamaguchi, A., Hojo, M., Kitamura, Y., Kameda, J., Ujiie, K., Hamada, Y., Hamahashi, M. and Hina, S., 2012. Tectonic mélangé as fault rock of subduction plate boundary. *Tectonophysics*, 568, pp.25-38.

Line 19: Consider replacing the sentence "However, LFEs and VLFs within..." with a sentence that better describes the problem investigated by the paper (that the field observations motivate a weak block as opposed strong block model).

Line 59: Nothing in the models reproduces tremor or LFEs. Suggest rewording to "shear failure that may correspond to tremor".

Line 85: In the "classic" model, as described here, block clogging is a state where the block density is relatively high, but in real shear zones, the rearrangement/redistribution of blocks would take geological time. This is not modeled in the literature cited here, so it is not fair to state that this model somehow contradicts the observations of LFEs or other slow slip phenomena based on issues of timescale.

Line 97: this is a spurious argument. Many geological materials can yield when the shear stress is 3 MPa, particularly clay-rich rocks such as dominate shallow subduction systems, particularly at low effective stresses characteristic of shallow subduction systems. What is the "yield strength" referred to here. Geological materials generally show a pressure-dependent yield behavior at low effective stress.

Line 98: The blocks and matrix in 2A have the same (high) cohesion: doesn't this show strong blocks in a strong matrix?

Line 111: I disagree with this sentence. The predicted structures that are outlined in this paragraph closely resemble those seen in subduction melanges such as the Mugi Melange. There is therefore good reason to think a strong block model is relevant. The size of the regions that yield can achieve the 10s m size stated in the text. Note that the blocks in any melange do not have constant size, they are well-documented to exhibit a power-law distribution of sizes (e.g. as defined by major axis lengths or apparent 2D areas). The size limiting mechanism cannot therefore be the size of the blocks, which seems to be what the authors propose here because they propose failure of the blocks is the source of LFEs. The geometric complexity of the shear network in the matrix seems a good candidate to me. I suggest deleting this sentence.

Line 205: This paragraph needs editing to improve the focus. The discussion of LFE source parameters is not matched by the discussion of the model. What "specifics" from the weak block scenario match the inferred source parameters of LFEs?

Line 241: As well as "deep tremor", the authors should compare their results to other well-described melanges exhumed from shallow subduction systems. The matrix composition of the Osa melange is atypical for subduction melanges, which more commonly have pelitic matrix. The "strength" of the matrix is important, so these other, more common rocks, should be discussed.

Line 246: This discussion mis-represents the conditions at which deep tremor occurs, which the Audet and Kim paper clearly shows is not exclusively low T, high P. Furthermore, I don't know of a single exposure of a high-grade rock that has serpentinite or blueschist pods/cores in an eclogite matrix. Perhaps the authors could cite a study that presents on here. This extrapolation of the model implications is too speculative.

Line 259: This sentence is unsubstantiated and should be reworded to acknowledge that this is a proposal/hypothesis rather than a concept that is proven by this study.

Below are our pPoint by point responses to editorial and reviewer comments. All comments are in black, with our responses in red.

REVIEWER COMMENTS (in black, with our responses in red)

Reviewer #1 (Remarks to the Author): (We have taken the comments from the reviewer's attached PDF file)

The response from the authors is detailed, and fully answers to almost all comments. I have a few additional small comments.

L 19-21, 38-40.

I am sorry but I have to confirm my comment, I will try to better detail. As you wrote, Bostock et al. (2015) showed a near-constant duration ($T \propto M_0^{1/10}$, very different from the regular scaling $T \propto M_0^{1/3}$) for LFEs in Cascadia. But this result is reported for LFEs, not VLFs. A near-constant duration for VLFs should mean a weak duration dependence upon moment, with a very small exponent (similarly to Bostock et al. 2015). In ref. 5-7 (now 6-8), I am not able to find a similar result. On the contrary (and similar to what you wrote in your answer!), the last lines of the abstract of ref. 8 seem to be a very good picture of the current literature about VLF scaling: "Given the non-uniqueness in possible source durations, we suggest it is premature to draw conclusions about VLF event sources or how they scale".

We think it best to simply add an additional figure (now Methods Figure 3) to the Methods section that shows a well-accepted summary of the observations by Gomberg et al. Versions of this figure have appeared in several review papers, which is why we had removed it when trying to shorten the MS for submission. This figure, to us, is a clear visual summary of what we are trying to point out about the relative 'near constant' duration of events over two orders of magnitude — and will let the reader form their own impression of what we state. (Note that reviewer 2 was the seismologist Gomberg who did not disagree with our previous first-order interpretation of the observations.). We are well aware that there is still an ongoing seismological debate as to whether the trends seen in LFEs are robust, and/or possibly variable between different regions with observed LFEs, and whether observed slopes are $T \propto M_0^{1/10}$, or $T \propto M_0^{1/3.5 \pm 0.5}$, etc., as these difficult measurements are at the current state-of-the-art in observational seismology. However, we feel this discussion would be beyond the scope of our MS, although we now mention it in an additional sentence place in the revised text. (W~~w~~e had already mentioned in one place in the methods section, but the reviewer seemed to not find this mention to give enough weight to this issue). Again, for our study, the only important observation is the non-controversial "relative near constant duration of events over two orders of magnitude" that we now further highlight in this revised text.

L 19-21, 38-40. I do agree with just describing the current seismic observations. However, seismic observations of Supino et al. (2020) show a self-similar moment-

duration scaling for LFEs in Nankai, which is the opposite of what you described. The simplest way to describe the current seismic observation could be that the LFE moment-duration scaling changes depending on the subduction zone analyzed.

You may choose not to consider the results of Supino et al. (2020). However, the results of Supino et al. (2020) are quite used in this manuscript.

We now further mention the current observational discrepancy between Supino et al.'s findings of LFEs scaling as $T \propto M_0^{1/3.5 \pm 0.5}$ in Japan vs. Bostock et al.'s findings of $T \propto M_0^{1/10}$ in Cascadia. The reader can judge for themselves what a $T \propto M_0^{1/3}$ or $T \propto M_0^{1/10}$ slope would look like on the new figure 3 ~~of~~ added to the methods section, and compare it to the summary of observations shown in that figure. Again, we do not wish to go into details of the current frontiers in seismic research on this issue, as for us this is a distraction from the focus of our paper.

L 40-41. Is Figure 1 showing information about the near-constant <1s duration of LFEs? We apologize if this is the reason that the reviewer found our prior remarks somewhat confusing. We now point the reader to the newly added figure 3 in the Methods section. That panel was once part of Figure 1 in the MS, then removed to save space, but we still referred to the missing panel as noticed by the reviewer. Thank you!

Reviewer #2 (Remarks to the Author):

The authors have done a fine job addressing the Reviewers' concerns. I think this paper is ready for publication. **Thank you!**

Reviewer #3 (Remarks to the Author):

Review of 'A strength inversion origin for non-volcanic tremor' by Vannucchi et al.

This is the second time I have reviewed this manuscript, previously as reviewer 3. In general, the authors have responded constructively to the previous round of comments from reviewers.

This study integrates numerical modeling and laboratory rock mechanics measurements to demonstrate that the fracture patterns within blocks in the Osa Melange, Costa Rica, formed under conditions where the blocks were relatively weak compared to the surrounding matrix.

The manuscript has two major problems. First, it sets out to disprove the "strong blocks" model.

Note: This misreading is not true, we absolutely do not dispute that in many melanges the blocks are stronger than the matrix, and are not trying to disprove the "strong blocks" scenario as the reviewer suggests. What we do dispute in this text is a different issue — whether the strong block model is the appropriate geomechanical scenario that leads to tectonic tremor. However, in the revision, as suggested by the editor, we somewhat tone down this issue to focus on the key new contribution of our paper — 'A strength inversion origin for non-volcanic tremor'.

However, the author's arguments about the mechanical considerations and geological predictions of the strong block model are inconsistent and without observational basis. The paper

would be stronger if it acknowledged the strong block model is a valid hypothesis. To address this misreading, we have further clarified that in many cases, melange blocks are stronger than their matrix, and further pointed out where (non-tremorgenic) numerical models with strong blocks generate structures that match the structures seen in some exhumed melanges.

Second, the manuscript attempts to portray the “weak block” model as applicable to LFEs generally. While the study results do a good job of explaining the Osa Melange features, I disagree that the weak block model can be generally representative. The volcanoclastic matrix of the Osa Melange is atypical. Subduction melanges exhumed from shallow depth more commonly contain a pelitic matrix. The fractures observed in the blocks of the Osa melange are not seen in other melanges (e.g. see the excellent map of the Mugi Melange in Kimura et al. (2012)). The key parameter that changes between the strong block/weak matrix and weak block/strong matrix is the cohesion of the two components. The cohesion of pelitic rocks is typically of the order of 1 MPa or less (e.g. see compilations in the text books by Zoback (Reservoir Geomechanics), Jaeger, Cook and Zimmerman (Fundamentals of Rock Mechanics)), so would presumably always be less than a lithified rock. In the revision, we now discuss melanges with a pelitic matrix in more detail. Again, we are not disagreeing with the field observations on the Mugi Melange as the reviewer seems to have inferred. In the revision, we now include this example where appropriate.

Overall, I think the strength inversion idea is interesting and justified for the case study site, but I think the significance of the study findings concerning the mechanics of LFEs is overstated in the text. Furthermore, none of the seismological characteristics of LFEs discussed in the manuscript seem to imply the weak block model is relevant or preferred. We disagree — as we discussed in the text, LFEs appear to have similar size slip areas and variable stress drops, and these features seem more consistent with a weak block scenario than a strong block scenario. We absolutely do not disagree with the reviewer that the Mugi Melange is a subduction channel mélange — where we differ is that we consider the further possibility that subduction channel melanges can be non-tremorgenic when they are deforming by-in a ‘stronger block in weaker matrix’ mode of deformation. Nonetheless, in the revision, we have toned down the strength of our conclusions regarding the issue that the strong block in weak matrix mode is likely to be non-tremorgenic, to further focus on the central issue implied by the title of our MS, ‘A strength inversion origin for non-volcanic tremor’.

I have highlighted below some areas where the work is incorrect or unsubstantiated. I recommend the authors modify the text to rectify the problems outlined above. I think this could be done with minor revisions to the text, although these revisions do need to address the core theme of the paper.

Kimura, G., Yamaguchi, A., Hojo, M., Kitamura, Y., Kameda, J., Ujiie, K., Hamada, Y., Hamahashi, M. and Hina, S., 2012. Tectonic mélange as fault rock of subduction plate boundary. *Tectonophysics*, 568, pp.25-38.

Line 19: Consider replacing the sentence “However, LFEs and VLFs within...” with a sentence that better describes the problem investigated by the paper (that the field observations motivate a weak block as opposed strong block model). We have kept this sentence, as the statement that the LFEs and VLFs have ‘nearly constant source durations for all observed magnitudes’ is a key observation for us that implies that these events are associated with failure of regions of near-constant size. However, we add a few lines later that the geological observations and samples for the lab experiments we perform all come from the Middle American Trench, to alert the reader that our observations are indeed focused on this region (a point the reviewer wanted us to emphasize).

Line 59: Nothing in the models reproduces tremor or LFEs. Suggest rewording to “shear failure that may correspond to tremor”. We had said ‘can lead to tectonic tremor’. In order to clarify that

it can lead to conditions that induce tremor, we have reworded this sentence to 'can lead to shear failure that generates tectonic tremor'

Line 85: In the "classic" model, as described here, block clogging is a state where the block density is relatively high, but in real shear zones, the rearrangement/redistribution of blocks would take geological time. This is not modeled in the literature cited here, so it is not fair to state that this model somehow contradicts the observations of LFEs or other slow slip phenomena based on issues of timescale.

We think the reviewer misunderstood our argument, which is specifically noting that the Beall et al. numerical experiments led them to propose that block clogging and breaking was associated with tremor, while periods without clogging would be associated with higher sliprate pulses that correspond to SSEs. We noted that tremor is correlated with SSEs, not anticorrelated as the Beall et al. hypothesis would predict. We have further clarified that we are specifically referring to the Beall et al. hypothesis in these lines.

Line 97: this is a spurious argument. Many geological materials can yield when the shear stress is 3 MPa, particularly clay-rich rocks such as dominate shallow subduction systems, particularly at low effective stresses characteristic of shallow subduction systems. The reviewer misunderstood what we were trying to show. We agree that several geological materials can yield when the shear stress is 3 MPa or less. This is why we had presented a scaling argument valid for any channel, and then used one set of 'typical' parameters. Now we explicitly also mention the parameters which would lead to order of magnitude lower channel stresses, too, to make it clearer that order of magnitude lower channel stresses are also geologically viable. What is the "yield strength" referred to here. Geological materials generally show a pressure-dependent yield behavior at low effective stress. This question is exactly why we had summarized different discipline's varying characterizations of 'strength' of materials in Table S1 of the supplementary material, and directly discussed this issue near the beginning of the previous paragraph, e.g. 'Note that in geology, *strength* is usually defined as the resistance to permanent deformation by *either flow or fracture*²⁴ (see Table S1 for common rheological terms in geology, and their corresponding terminology in continuum and fracture mechanics).'

Line 98: The blocks and matrix in 2A have the same (high) cohesion: doesn't this show strong blocks in a strong matrix? We have tried to be very accurate in how we define the rheologies studied here, distinguishing between viscosity and cohesion (note that field geology terminology often equates higher viscosity with higher strength). Note there is a table in the beginning of the supplement that addresses this point in a clear way, and this table is cited in our introduction to the place where we precisely define the rheological terminology used in this paper (see previous response).

Line 111: I disagree with this sentence. The predicted structures that are outlined in this paragraph closely resemble those seen in subduction melanges such as the Muji Melange. We agree that the models discussed here predict structures similar to natural examples. This was why we noted how the structures produced in scenarios in Fig 2B and Fig 2C correspond to the known geologic structures that we cited, and now add a further reference to the Muji Melange example highlighted by the reviewer. However, the models in Fig 2B and Fig 2C do not develop failure modes consistent with the characteristic tens of meters failure size and variable yield stress characteristics of observed seismic tremor — that is the additional point that we made here, that these mélange structures may *not* be a good proxy for places where the strong blocks in these melanges are tremorgenic asperities. We now 'tone down' our emphasis on this point by removing the previous ending to this section 'and that using strong blocks as a geological proxy for tremorgenic "asperities" is incorrect' and that using strong blocks as a geological proxy for tremorgenic "asperities" is likely to be incorrect, and that using strong blocks as a geological proxy for tremorgenic "asperities" is likely to be incorrect, and that using strong blocks as a

Comment [MJ1]: Add this example?!

~~geological proxy for tremorgenic “asperities” is likely to be incorrect, and that using strong blocks as a geological proxy for tremorgenic “asperities” is likely to be incorrect. summary sentence by stating ‘likely to be incorrect’ instead of ‘is incorrect’.~~

There is therefore good reason to think a strong block model is relevant. The size of the regions that yield can achieve the 10s m size stated in the text. Note that the blocks in any melange do not have constant size, they are well-documented to exhibit a power-law distribution of sizes (e.g. as defined by major axis lengths or apparent 2D areas). The size limiting mechanism cannot therefore be the size of the blocks, which seems to be what the authors propose here because they propose failure of the blocks is the source of LFEs. The geometric complexity of the shear network in the matrix seems a good candidate to me. I suggest deleting this sentence. **The reviewer misuses the finding that blocks in natural mélanges typically have a power-law distribution of sizes, because there is always a maximum size where this power-law relationship stops. The key length-scale, in our opinion, is exactly the maximum size where the power-law distribution breaks down — i.e. the initial sizes of the blocks. This largest characteristic size is what we refer to in the text. We added a sentence in the discussion to further clarify this point.**

Line 205: This paragraph needs editing to improve the focus. The discussion of LFE source parameters is not matched by the discussion of the model. What “specifics” from the weak block scenario match the inferred source parameters of LFEs? **The “specifics” that match the inferred source parameters are the inferred stress-drops and durations (linked to characteristic slip sizes) of the LFEs, as already noted in the text.**

Line 241: As well as “deep tremor”, the authors should compare their results to other well-described melanges exhumed from shallow subduction systems. The matrix composition of the Osa melange is atypical for subduction melanges, which more commonly have pelitic matrix. The “strength” of the matrix is important, so these other, more common rocks, should be discussed. **We now discuss this point in the discussion.**

Line 246: This discussion mis-represents the conditions at which deep tremor occurs, which the Audet and Kim paper clearly shows is not exclusively low T, high P. Furthermore, I don’t know of a single exposure of a high-grade rock that has serpentinite or blueschist pods/cores in an eclogite matrix. Perhaps the authors could cite a study that presents on here. This extrapolation of the model implications is too speculative. **We have rephased the sentences in question to “For example, during prograde metamorphism, heterogeneous phase transformations would reflect spatial variations in net fluid/rock ratio and/or protolith composition. Transformations of blueschists to eclogites^{23,57} and/or ultramafic phases to serpentinites⁵⁸ would generate a strength inversion of ‘blocks’ and ‘matrix’ mechanically analogous to what we propose to sometimes occur along a shallow megathrust.” We have now also added further references that describe eclogitization linked to devolatilization of adjacent materials, and now better clarify this point at the cost of more words and two added references.**

Line 259: This sentence is unsubstantiated and should be reworded to acknowledge that this is a proposal/hypothesis rather than a concept that is proven by this study. **Here and elsewhere in the MS, we have ‘toned down’ our wording to acknowledge that this is a proposal/hypothesis that is motivated and supported by the evidence presented in this MS.**

REVIEWERS' COMMENTS

Reviewer #1

1. Line 19, 111, 205.

It could be possible to improve the manuscript as suggested by these points. I think there is a common misunderstanding caused by discussing in the main text the "characteristic size" of LFE source without specifying that what the authors mean is a "largest characteristic size" (as better explained replying to one comment). This is actually a strong seismological characterization of LFEs (the maximum seismic moment (magnitude) observed for this class of events is $\sim 10^{13}$ N m for almost all subduction zones) while the source scaling is more controversial.

Probably, this would also help to reconcile the disagreement about the fact that the strong block model is/is not compatible with seismological observations. For instance, looking at reference 25, this model seems to be compatible with a maximum length scale for LFEs (if I am correct, this was what reviewer 3 was suggesting).

2. "Overall, I think the strength inversion idea is interesting and justified for the case study site, but I think the significance of the study findings concerning the mechanics of LFEs is overstated in the text. Furthermore, none of the seismological characteristics of LFEs discussed in the manuscript seem to imply the weak block model is relevant or preferred." and " Line 259: This sentence is unsubstantiated and should be reworded to acknowledge that this is a proposal/hypothesis rather than a concept that is proven by this study."

Also considering my previous comment, I tend to agree here with the reviewer and probably the discussion could be improved in terms of challenging questions raised by this study.

Reviewer #4 (Remarks to the Author):

Dear authors,

I have read with great interest your contribution. From the documents that I have been given access to, I gather that this is the third round of review that your manuscript goes through. Speaking for my own experience, this is indeed frustrating and I am myself little clear as to why a manuscript should go so many times out to reviewers, who, by the way, also change during the process. It is obvious to us all that the more a manuscript is seen by different referees, the more inputs, criticisms and opinions it receives. To make things worse, moreover, it is common for different reviews to significantly diverge, thus making the life of authors quite frustrating.

I do wish for more decisions by the editors, who, after two full rounds of revisions, should certainly have had enough inputs and elements to make up their minds as to the overall quality of the submitted work and whether it can be accepted.

I worked on this manuscript having this as background in my mind. In summary, after reading the rebuttal, I abstained from an overly detailed review and tried instead to gather a general impression by also assessing how the initial criticisms have been dealt with.

This is an interesting, novel and thought-provoking paper that deserves publication. Its content, the dataset it's based on, the conceptualization and the writing style all comply with the level expected for a journal of the Nature series.

I attach a pdf with a few comments, suggestions and a couple of requests for clarification. I am sure the authors can easily address all of those.

The one significant point that I raise is perhaps the need to better define the temporal evolution of the "weakening mechanisms" leading to the inverted rheology in the context of subduction. In particular, Figure

3 F is sort of very little helpful at the moment. It is not clear, it expresses no time dimension, nor it is possible to connect the illustrated fracturing of the blocks to any progressive deformation story. That I found quite frustrating, as there is no figure elaborating upon a summary conceptual model to be used to merge the data, the results as well as the numerical models into a synthesis (with iconographic support). I read that the authors have added Fig. 3 to the paper in response to a point made by one of the reviewers. I honestly do not understand the need for that figure (it is sufficient to read the literature on the duration of the sources...), while I do miss a figure presenting a model where everything falls into place. To clarify what I have in mind, I refer the authors to Fig. 10 of the paper by Braden and Behr (2021, JGR), which studies in detail the mechanisms of basalt weakening during subduction in a channel. Admittedly, the authors of that paper conclude "strong block in weak matrix", but that figure does make a difference while reading their work.

I think that this study would also greatly benefit from a summary figure.

A second point that could possibly require some improvement is the characterization of the "melange", which is not that well described, particularly for the volcanoclastic matrix. What about much more common pelitic matrix material? How does this peculiar composition (which requires some further details on the genesis of the rock) potentially affect the system as a whole?

Lastly, I suggest that a couple of extra sentences be added as to the representativeness of the proposed model. This has to be necessarily connected with the efficiency of the processes that are capable to lead to the observed rheology inversion. The authors themselves write about a patchy distribution of this inversion, if the cause thereof is to be sought in the zeolitization process of the matrix. How representative can this be? Can this be discussed in terms of spatial and temporal implications? Is it appropriate to reconsider the quite well-established concept of hard-block in weak-matrix on the basis of some potentially "only patchy" process? How impacting can this be not at the scale of the sampled Osa Melange, but at that of the entire subduction channel? This is intriguing and highly fascinating and I think that a short discussion deserves to be added to the text.

A few other specific aspects are referred to directly in the annotated pdf and I would ask the authors to take a look and see whether they agree with my take.

Yours sincerely

Giulio Viola

REVIEWER COMMENTS (in black, with our responses in red)

Reviewer #1

Reviewer #1

1. Line 19, 111, 205.

It could be possible to improve the manuscript as suggested by these points. I think there is a common misunderstanding caused by discussing in the main text the "characteristic size" of LFE source without specifying that what the authors mean is a "largest characteristic size" (as better explained replying to one comment)...

We have now modified Line 21 and Line 119 to add the word 'characteristic' (e.g. 'characteristic size')

2. [Reviewer 1 quoting reviewer 3] "Overall, I think the strength inversion idea is interesting and justified for the case study site, but I think the significance of the study findings concerning the mechanics of LFEs is overstated in the text. Furthermore, none of the seismological characteristics of LFEs discussed in the manuscript seem to imply the weak block model is relevant or preferred." and " Line 259: This sentence is unsubstantiated and should be reworded to acknowledge that this is a proposal/hypothesis rather than a concept that is proven by this study."

Also considering my previous comment, I tend to agree here with the reviewer and probably the discussion could be improved in terms of challenging questions raised by this study.

We have tried to further clarify the discussion to address this point.

Reviewer #4 (Remarks to the Author):

...

This is an interesting, novel and thought-provoking paper that deserves publication. Its content, the dataset it's based on, the conceptualization and the writing style all comply with the level expected for a journal of the Nature series.

I attach a pdf with a few comments, suggestions and a couple of requests for clarification. I am sure the authors can easily address all of those.

The one significant point that I raise is perhaps the need to better define the temporal evolution of the "weakening mechanisms" leading to the inverted rheology in the context of subduction. In particular, Figure 3 F is sort of very little helpful at the moment. It is not clear, it expresses no time dimension, nor it is possible to connect the illustrated fracturing of the blocks to any progressive deformation story. That I found quite frustrating, as there is no figure elaborating upon a summary conceptual model to be used to merge the data, the results as well as the numerical models into a synthesis (with iconographic support). I read that the authors have added Fig. 3 to the paper in response to a point made by one of the reviewers. I honestly do not understand the need for that figure (it is sufficient to read the literature on the duration of the sources...), while I do miss a figure presenting a model where everything falls into place.

To clarify what I have in mind, I refer the authors to Fig. 10 of the paper by Braden and Behr (2021, JGR), which studies in detail the mechanisms of basalt weakening during subduction in a channel. Admittedly, the authors of that paper conclude "strong block in weak matrix", but that figure does

make a difference while reading their work.
I think that this study would also greatly benefit from a summary figure.

Thank you for your suggestion. We have now added a time-evolution conceptual model to the paper as Figure 6 that follows the style of the summary figure in Braden and Behr.

A second point that could possibly require some improvement is the characterization of the "melange", which is not that well described, particularly for the volcanoclastic matrix. What about much more common pelitic matrix material? How does this peculiar composition (which requires some further details on the genesis of the rock) potentially affect the system as a whole?

The detailed characterization of the Osa Melange is given in the supplementary material. However, We have also extended the description of this *mélange* in the text. Please note that this composition is not peculiar, especially for material on the seafloor, and volcanoclastic material makes up about 40% of the total input into subduction zones according to Plank (2013), which we now cite. We also now further discuss the similar implications for other possible *mélange* compositions. (also addressing the next point raised by the reviewer.)

Lastly, I suggest that a couple of extra sentences be added as to the representativeness of the proposed model. This has to be necessarily connected with the efficiency of the processes that are capable to lead to the observed rheology inversion. The authors themselves write about a patchy distribution of this inversion, if the cause thereof is to be sought in the zeolitization process of the matrix. How representative can this be? Can this be discussed in terms of spatial and temporal implications? Is it appropriate to reconsider the quite well-established concept of hard-block in weak-matrix on the basis of some potentially "only patchy" process? How impacting can this be not at the scale of the sampled Osa Melange, but at that of the entire subduction channel? This is intriguing and highly fascinating and I think that a short discussion deserves to be added to the text.

We have extended the discussion to further address this point. (Also see our previous response).

A few other specific aspects are referred to directly in the annotated pdf and I would ask the authors to take a look and see whether they agree with my take.

Responses to the comments on the annotated PDF

Line 37 – although studies of the rock record also suggest possible hybrid (mixed-mode) failure, with Mode I opening associated with localised shear (Fagereng et al, many times, Uije et al., 2018; Giuntoli and Viola 2021).

We now add the word 'seismic' to make it clearer that tremor events are visible as seismic shear failure events. We think the intro is already quite long, and feel it is best to keep this part of the intro focused on the seismic characteristics of tremor. We discuss implications drawn from the rock-record later in the paper (e.g. lines 81-90 of the revised text where we specifically mention this point).

Line 57. In this short summary I think it would be useful to provide a simple synthesis of what is the proposed "geological" record of tremor, which may help strengthen the objectively difficult to understand "transition" from the geophysical record to the physical expression of the possible sources.

Hydroshears are indeed good candidates, thus not only "asperities".

This is an important element to factor in because it relates to the faith of existing features (indeed, the asperities) vs. newly created objects (mixed-mode fractures) as the responsible for the source of tremors.

Kirkpatrick et al (2021) provide a useful review, but also Behr and Burgmann 2021. Besides, I think it may also be beneficial to the actual discussion of the paper and the proposition of the authors' new take on the "broken formation" rheology in the subduction channel.

I think that this intro could easily be "lengthened" and would thus become more beneficial to the reader.

In this paper, we don't go into so much detail on the geological record of tremor because we focus on the evidence for the weak block in strong matrix transition that we propose, and the geological evidence that this transition can and has taken place within a subduction shear channel. The reviewer suggests that we should focus more on the geological record of mixed-mode failure in subduction shear zones. We are well aware of this very interesting phenomena and have even written papers presenting field evidence for this (e.g. Vannucchi et al., 2010), but feel this topic, while highly interesting, is a bit of a distraction to the main topic of this paper. In short, we think that broadening the introduction to include the geological record in depth (with it not being used in any later arguments in this paper), would not be the best structure for this short paper.

Line 63 can it? by doing what? Even in low- geothermal gradient subduction zones, what would be the effects of reaching up to 1 GPa (and more) of environmental pressure?

Either elaborate further on this or be more dubitative as to the validity of a straight extrapolation.

We further elaborate here that this behaviour could occur in the P-T interval where the basalt-to-eclogite transition is generally presumed to take place, e.g. that metamorphic phase transitions of block w.r.t. matrix can lead to changes in relative strength that induce the onset of tremorigenic failure. This point is discussed at more length later in the discussion and is also discussed in our (new) conceptual summary figure suggested by the reviewer (thank you!).

Line 68 Replace often by commonly. **Done.**

Line 71 add 'discrete'. **Done.**

Line 71 change 'subdomains' to something better implying a rock volume. **Done.**

Line 81 delete 'clog'. **We keep clog as we think it helps (Note that this parenthetical comment was added in response to a prior reviewer comment)**

Line 87 "Contradict" in my humble opinion is not the right word. If anything, this conceptual model is not "compatible" with the observation, thus requiring further thinking. An observation is "made", its object is seen and it simply exists. We cannot avoid this. **Done.** ('to contradict' is now changed to 'incompatible with')

Line 89 I would suggest the authors use a sentence or two to further elaborate this point, as this is indeed crucial to the essence of the entire paper. Why is it not compatible in their opinion?

What would be, instead, a first-order observation that would satisfy "the stronger block in weaker matrix model"?

I think this aspect requires clarification to make sure that the reader follows their logics when then elaborating their study.

I suppose that the "not anticorrelated as posited" is not too clear as it stands at the moment in the syntax of the sentence and in the framework of the reasoning by the authors.. **We now have elaborated this to further clarify this point.**

Line 97. or shear zone? **Done.** We rephrased to 'subduction shear zone'

Line 98. Remove comma. **Done.**

Line 103. [Clarify] which of the two end-member cases, exactly? **Done.**

Line 111. Are you not discussing brittle failure of the matrix, as stated in the line above? In that case, shear zone is certainly not the right term to refer to the effects of localization therein. **Changed.**

Line 128. Can you provide the actual [igneous] rock type end-members? **Briefly stated. Also note that we have just pointed the reader to the Supplementary Information to see more in-depth information.**

Line 128. excuse my ignorance, but what do you mean by "pelagic block"? Can you stick to a rock type that the reader can more easily associate some mechanical parameters with? **Clarified.**

Line 130. [again further clarify rock descriptions. **Done.**

Lines 131-132 Remove plurals. **Done.**

Line 144. what do you mean? the faulted/brecciated individual blocks or the entire melange package now? There is a dimensional jump in your description or are you still at the evidence of Fig. 3? By the way, please see Fig. 3 for some further comments. **Clarified.**

Line 151. This shall not be taken for granted. I assume it is in the interest of the authors to better elaborate this and, at the cost of being redundant, to remind the reader why

they conclude this. This was further elaborated in the next section, as realized and noted by the reviewer later.

Line 153. One aspect that shall at least be mentioned I guess is the timing of deformation. Can it be firmly concluded when the blocks got fractured? Are there indications, for example, that no fracturing occurred during the exhumation and therefore under retrogressive conditions?

What we see in Fig. 3 has to be implemented by some suggestions as to the time scale in terms of progressive fracture evolution. Else the risk arises to only consider the overall summation of deformation increments while ignoring whether the various increments all represent the same event.

The same applies to the timing of mineralogical changes/weathering/alteration. This was further elaborated later in this section, as realized and noted by the reviewer later.

Line 159. I see here at you address here some of my inputs above. Very good. Thanks! – we felt this was the best order to address these issues.

Line 167. Excellent! Thanks! (This was where we further addressed the reviewer's previous questions/suggestions.)

Line 180. keep it to the context of the experiment: deformation. I am just saying that here you describe the experiment. Extrapolations will come later. We are describing the behaviour seen in the numerical experiment. We removed the word 'channel' in case this was what was confusing the reviewer to think we were talking about more general subduction shear channel behaviour.

Line 195. Reading this, it becomes truly necessary to draw a parallel to the work of Braden and Behr (2021, JGR), at least for the geometrical similarities and the envisaged process, which is, indeed, quite similar, except for the final "end result"... See their Figure 10. Note that Braden and Behr were familiar with our work by the time they wrote their paper (our paper has been 3 years in review in Nature Geoscience and then Nature Communications), and Whitney Behr had even invited Vannucchi to ETH to present and discuss our hypothesis. Braden and Behr are not talking about our proposed mechanism, so we decided not to add this reference. To draw a parallel would indeed be interesting in a longer expanded discussion of subduction shear-zone deformation mechanisms.

Line 230. This is indeed interesting to reflect upon: how pervasive would be the potential processes leading to rheology inversion? Hydrothermalism is known to be very heterogeneous of a process, affecting rock volumes that potentially can be quite different in size and in a very unpredictable, patchy manner. Would zeolitization, for example, be sufficient to cause "enough inversion" to create a dominant rheological style in the way you envisage it in a subduction channel?? This is indeed what we suggest here, and then point out some evidence that currently supports this interpretation. Obviously this is still a hypothesis that needs further testing.

Line 309 (Table 1) Why are these values identical for both the matrix and the inclusions? The way it is presented, the table makes little sense. The viscosity is obviously the parameter that creates differences between matrix and clasts. It is perhaps more meaningful to chart the actual conditions of each experiment instead of listing the end-member values of all parameters, which, as shown here, generates a rather useless table. Besides, is it really realistic to have identical densities between matrix and inclusion? **We tried to use the approach of keeping the numerical experiments 'as simple as possible, but no simpler'. That is why we kept as many parameters the same as possible between blocks and matrix, varying only their cohesion and viscosity.**

Line 374 (Caption Figure 2) what cores? cores of what? Please, explain better. **Done.**

Line 511. (Figure 3 caption) This sentence is more appropriate for figure F, where you provide a scheme for the progressive evolution of the block in a time-integrated model. **Moved. We have also added a further panel showing evolution instead of a snapshot of evolution.**

Line 513. Honestly, I do not find 3F particularly useful or informative in the current configuration. What is the difference between the first block on the left and the last on the right? Is there a progressive deformation (time somehow involved) or is it a snapshot at any moment during deformation? If the latter, why do you have 5 blocks that to a first glance do not seem to differ from each other for any specific feature? **We were not trying to show time evolution in 3F, but rather to show an idealization of the model calculation state and why it was consistent with the 'final' images seen in the geological field evidence shown in panels 3D/3E. We now add an idealized summary conceptual model as Figure 6 that shows the anticipated space-time evolution proposed in our hypothesis for both shallow and deep subduction-channel-related tremor.**